# Chemotherapy-induced adipo-lineage cell senescence drives bone loss

Ganesh Kumar Raut[1], Taylor Malachowski[1], Anupama Melam [1],
Renata Ramalho-Oliveira[1], Taylor Holt[2], Xianmin Luo[1], Zhangting Yao[1],
Douglas V. Faget [1,3], Qihao Ren [1,4], David G. DeNardo [5,6,7] &
Sheila A. Stewart [1,5,6,7] ✉

Chemotherapy-induced bone loss is a debilitating and common side effect of cancer treatment, though its underlying mechanisms remain poorly understood. Here, we show that, despite the systemic administration of chemotherapy, cellular senescence is restricted to bone marrow adipo-lineage cells specifically Cxcl12-abundant reticular (CAR) cells and bone marrow adipocytes (BMAds). Induction of senescence within these populations promotes RANK ligand (RANKL)-mediated osteoclastogenesis, leading to significant bone loss. Notably, we find that inhibition of the p38MAPK-MK2 pathway suppresses the senescence-associated secretory phenotype (SASP), including RANKL production abrogating bone loss. Furthermore, treatment with the senolytic combination dasatinib and quercetin (D + Q) selectively eliminates senescent CAR cells and BMAds, effectively preventing chemotherapy-induced bone loss. Given that nearly all chemotherapy treated patients experience bone loss and associated fracture risk, our findings offer a promising therapeutic avenue to preserve bone integrity and improve quality of life for cancer patients.

Chemotherapy remains a major weapon in our anti-cancer armamentarium. However, while chemotherapy can slow tumor progression and in some instances be curative, it also drives a wide range of comorbidities that can significantly impact a patient's quality of life and in some instances, be so severe that it results in dose de-escalation that negatively impacts its anti-tumor properties and patient survival[1,2]. One such side effect is bone loss, which occurs in nearly all patients treated with chemotherapy. Patients who experience substantial bone loss are more susceptible to pain and fracture and patients that experience a fracture have reduced overall survival rates[3].

Senescent cells undergo cell cycle arrest and typically express p16, senescence-associated-β-galactosidase (SA-β-gal), and senescence associated secretory phenotype (SASP) factors that are post-transcriptionally regulated by the p38 mitogen-activated protein

kinase-MAPK-activated protein kinase 2 (p38MAPK-MK2) pathway[4]. The SASP consists of a plethora of cytokines, chemokines, growth factors and factors that remodel the extracellular matrix (ECM) and impact numerous cells including immune cells[5]. We previously reported that the elimination of senescent cells or inhibition of SASP production using genetic and pharmacological approaches to inhibit the p38MAPK-MK2 pathway prevented chemotherapy-induced bone loss[6]. However, the cells and mechanisms responsible remained elusive. Given a growing number of reports that the cell type undergoing senescence expresses unique SASP factors that can have a profound and specific impact on tissue homeostasis[7], our previous report left a number of critically unanswered questions. Further, the type of experienced stress can also dictate what cell types enter senescence. Thus, it was interesting to note that in the context of aging, senescent

[1]Department of Cell Biology and Physiology, Washington University School of Medicine, St. Louis, MO, USA. [2]College of Arts and Sciences, Washington University, St. Louis, MO, USA. [3]Pheast Therapeutics, Palo Alto, CA, USA. [4]Discovery Oncology, Genentech, South San Francisco, CA, USA. [5]Department of Medicine, Washington University School of Medicine, St. Louis, MO, USA. [6]Siteman Cancer Center, Washington University School of Medicine, St. Louis, MO, USA. [7]ICCE Institute, Washington University School of Medicine, St. Louis, MO, USA. ✉e-mail: sheila.stewart@wustl.edu

osteocytes were found to contribute to bone loss[8], and our studies failed to implicate osteocytes in chemotherapy-induced bone loss[6].

Using genetic and pharmacologic approaches, we found that chemotherapy-induced senescence was limited to bone resident CAR cells (senCARs) and bone marrow adipocytes (senBMAds) and these cells uniquely contributed to bone loss through increased expression of RANKL that led to increases in osteoclasts. Further, we found that osteoblast differentiation and mineralization capacity was reduced following chemotherapy. Thus, senCAR/senBMAds effectively decoupled bone formation and bone resorption, which are normally tightly linked, driving significant bone loss. Pharmacologic or genetic elimination of senCARs and senBMAds prevented chemotherapy-induced bone loss without impacting the anti-tumor activities of chemotherapy. Similarly, inhibition of the SASP protected from chemotherapy-induced bone loss, providing an orthogonal therapeutic approach. Together, these data raise the possibility that our approach could benefit patients experiencing chemotherapy-induced bone loss.

## Results

### Chemotherapy induces senescence and impacts bone homeostasis

Previously we demonstrated that several weekly doses of doxorubicin led to senescence and bone loss[6]. While the long-term damaging effects of chemotherapy on bone health are well known[9], it is unclear whether a single dose of chemotherapy is sufficient to induce senescence and drive bone loss. To address this question, we established a 9-day model (Fig. S1A). 12-week-old female mice (C57BL/6) were treated with either doxorubicin (Doxo) or Paclitaxel (PTX) and we used 3D μCT to examine bone density as measured by bone volume over total volume (BV/TV) on day 9. We found that a single 5 mg/kg dose of Doxo was sufficient to cause significant bone loss (Fig. S1B, C). In addition to bone loss, other bone parameters were also disrupted including a drastic reduction in trabecular number and thickness and increases in trabecular spacing was noted in chemotherapy-treated mice compared to vehicle (Veh) treated mice with unchanged cortical bone thickness (Fig. S1D–S1I). Similar results were observed when mice received clinically relevant PTX (Fig. S1J–S1Q). These findings were not limited to female mice, Doxo also induced bone loss in male mice (Fig. S1R–S1Y). Because bone volume is maintained by balanced osteoclast and osteoblast activity, we also interrogated them. Tartrate resistant acid phosphatase (TRAP), a marker of osteoclasts, staining on the femur of Doxo-treated mice showed significantly more differentiated osteoclasts that covered more bone surface (Fig. S2A–S2C). Using the Col1a1-GFP mouse that allows one to directly measure osteoblasts on the bone surface, we also found that osteoblast numbers were reduced in Doxo-treated mice (Fig. S2D, S2E) compared to Veh-treated mice. To assess osteoblast activity, we treated mice with calcein and alizarin red to assess bone formation. Using this strategy, we found that the bone formation rate (BFR/BS) was significantly decreased in Doxo-treated mice relative to Veh-treated mice after 9 days (Fig. S2F, S2G). Because the bone mineralization rate is dependent on collagen deposition, we also assessed collagen type-1 (Col1a1) production by quantitative RT-qPCR in crushed bone from which bone marrow was largely removed by centrifugation (hereafter termed bone-resident fraction), which is enriched in osteoblasts. We found that Col1a1 was significantly reduced upon Doxo treatment (Fig. S2H), raising the possibility that its reduction contributed to reduced mineralization rate. Collectively, these results demonstrated that a single dose of chemotherapy is sufficient to induce rapid bone loss and shifts the balance of bone remodeling toward bone resorption at the expense of bone formation.

To determine if senescence contributed to the rapid bone loss observed after a single dose of chemotherapy, we treated 12-week-old INK-ATTAC mice, that express a p16[INK4a] dependent inducible suicide gene[10] with Doxo followed by Veh or AP20187 (AP, 10 mg/kg) on days 2, 4, 6, and 8 to eliminate p16[+] senescent cells (Fig. 1A). To determine if senescence was induced in this time frame and if AP could eliminate p16[+] senescent cells, we performed senescence-associated beta-galactosidase (SA-β-gal) and p16 protein staining on treated bones and found that both were increased upon Doxo treatment and eliminated in the femurs of mice treated with AP (Fig. 1B–E). To ensure specificity of our p16 antibody we also stained bones from p16 knockout animals treated with Doxo and failed to detect p16 protein (Fig. S2I). We also evaluated Cdkn2a (p16) gene expression in the Doxo-treated bone-resident fraction. We found elevated levels of Cdkn2a mRNA in the bone-resident fraction that was abolished by AP treatment (Fig. 1F). To ascertain the impact of a single dose of chemotherapy on bone parameters, we subjected the femurs to μCT. We found that AP treatment of Doxo-treated mice restored trabecular bone density (Fig. 1G-I), bone mineral density (BMD) and trabecular thickness (Tb.Th) without significantly impacting other bone parameters (Fig. S3A–E). Similar results were obtained with PTX treatment (Fig. S3F–K).

Bone loss can result from increased osteoclastogenesis and/or decreased osteoblastogenesis. To ask if senescence impacted osteoclasts, we stained the femurs of treated mice for TRAP. As expected, we found that Doxo treatment increased osteoclast numbers compared to Veh-treated mice. When we examined the femurs of INK-ATTAC mice treated with Doxo+AP, we found that osteoclast numbers and coverage of bone were similar to Veh-treated mice (Fig. 1J–L). Next, we assessed the impact of chemotherapy on osteoblast numbers. For these analyses, we utilized osteocalcin staining (OCN[+] cells) to quantify osteoblasts. Femurs were isolated and sections stained with OCN. When we quantitated OCN[+] cell numbers on trabecular bone, we found they were decreased in the femurs from Doxo-treated mice and this was reversed upon AP treatment (Fig. S3L, M). We also assessed bone formation rates by treating INK-ATTAC mice with calcein and alizarin to assess bone formation rates and we found that the reduction in bone formation noted in Doxo-treated mice was restored by AP treatment (Fig. S3N, O). Taken together, these findings indicate that chemotherapy-induced senescence is associated with rapid disrupted bone homeostasis that leads to bone loss.

### Senescent bone resident cells drive chemotherapy-induced bone loss

Chemotherapy is a systemic treatment that impacts cells across the body, and bone homeostasis can be impacted by local bone resident cells and systemic signaling pathways[11]. To ask if bone resident cells drove bone loss, we employed a vossicle model system that allows one to transplant vertebral bodies from one syngeneic mouse into a recipient mouse allowing us to mix genotypes[12,13]. For these studies, the L4 and L5 vertebrae from 4-day-old neonate INK-ATTAC mice were implanted in wild-type 10-week-old C57BL/6 syngeneic recipient mice to ask if senescent bone resident cells were responsible for chemotherapy-induced bone loss. Two weeks after the implantation of the vertebral bodies, when a blood supply was established[14], mice were injected with 5 mg/kg doxorubicin (Doxo) weekly for 4 weeks (Fig. 2A, B). To assess bone loss, the vertebral bodies and femurs were removed from the mice 9 days after the final Doxo dose and bone density was assessed by μCT. Analyses of the vertebrate by μCT revealed that AP treatment prevented Doxo-induced bone loss in implanted INK-ATTAC vertebral bodies (L4 & L5) (Fig. 2C and S4A). In contrast, bone loss in recipient wildtype mouse femurs was not rescued by AP treatment (Fig. 2D). Furthermore, quantification of TRAP[+] osteoclasts in the vertebral bodies revealed that AP treatment also abrogated the increased osteoclast numbers in INK-ATTAC vertebrate (Fig. 2E–G, S4B–D) but failed to reduced TRAP[+] osteoclasts in recipient wildtype mouse femurs (Fig. S4E–G). Taken together, these findings indicate that senescent bone resident cells drive chemotherapy-induced bone loss.

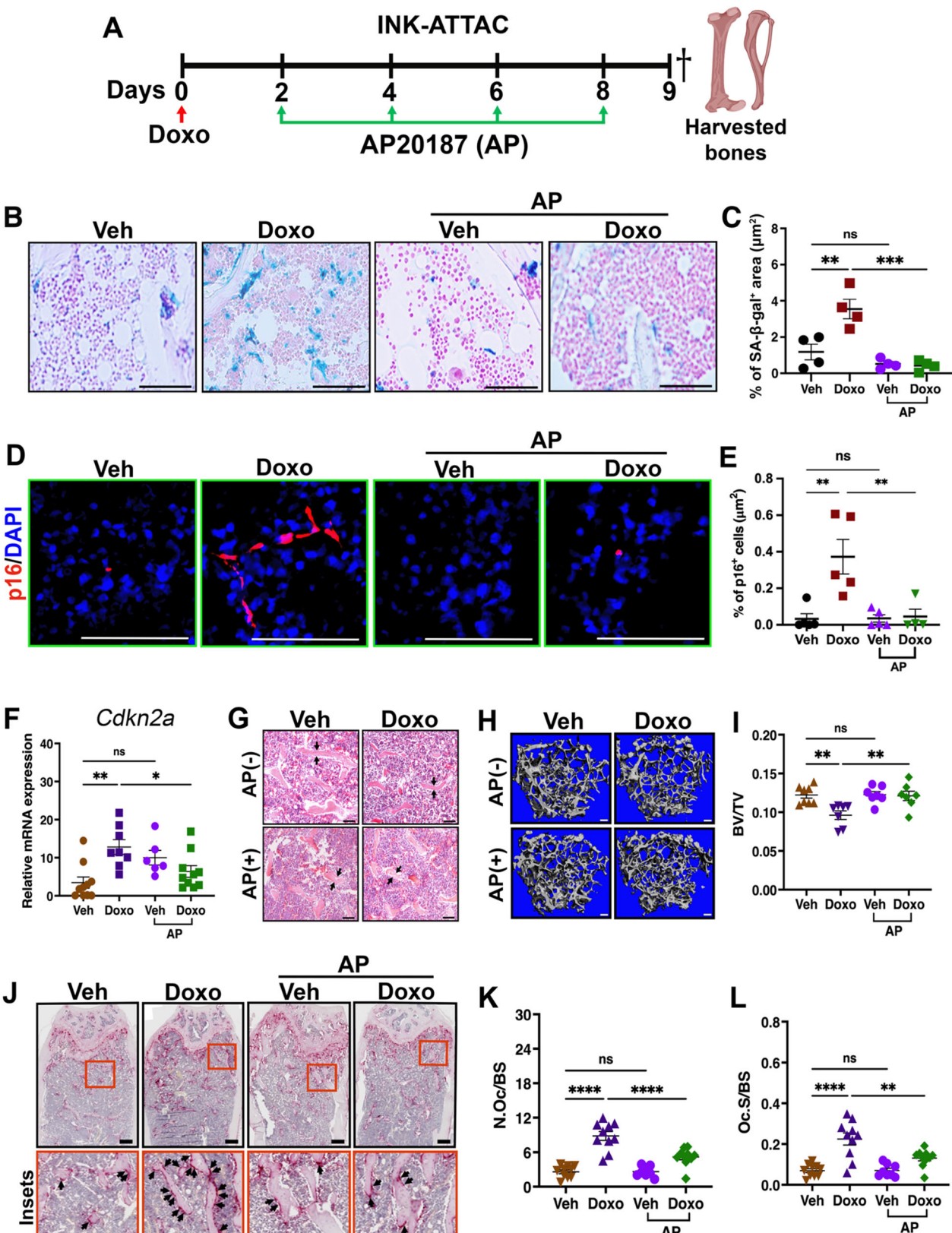

Vertebrate contain CD45⁺ stem cells and immune cells including T cells and macrophages, which can contribute to bone homeostasis[15] and can undergo senescence[16,17]. Thus, we next asked if senescent CD45⁺ cells contributed to chemotherapy-induced bone loss by carrying out a bone marrow transplant using bone marrow from INK-ATTAC mice into wildtype mice. This allowed us to induce the *Cdkn2a*-

driven suicide gene in senescent CD45⁺ derived cells exclusively. Recipient wildtype C57BL/6 (CD45.1) mice were treated with two doses of 400 cGy 4 hours apart and then received bone marrow (5 × 10⁶ cells/200 µl/mouse) from INK-ATTAC mice (CD45.2) and animals were allowed to recover for 6 weeks (Fig. S5A). To assess chimerism, peripheral blood mononuclear cells (PBMC) from recipient mice were

**Fig. 1 | Chemotherapy induces senescence and impacts bone homeostasis.**
**A** Schematic of the experimental timeline for dosing regimen for doxorubicin (Doxo) and AP20187 (AP) in 12-week-old INK-ATTAC mice. Dagger indicates time of sacrifice and bone harvest. Schematic was created in BioRender. Stewart, S. (2025) https://BioRender.com/71wz45j. **B, C** Representative images and quantification of SA-β-gal$^+$ area (blue). Scale bar: 100µm. $n = 4$/group. **$P = 0.0026$; ***$P = 0.0002$; $P = 0.5700$; ns = not significant. **D, E** Immunofluorescence (IF) staining of femur sections and quantification of p16$^+$ (red) area. Scale bar: 50µm. $n = 5$ mice (Veh), 5 mice (Doxo), 5 mice (Veh+AP), and 4 mice (Doxo+AP). **$P = 0.0026$; **$P = 0.0057$; $P > 0.9999$, ns = not significant. **F** Cdkn2a (p16) expression level in the bone-resident fraction by RT-qPCR. Actin and cyclophilin were used as housekeeping genes. $n = 10$ mice (Veh), 8 mice (Doxo), 6 mice (Veh+AP), and 10 mice (Doxo+AP). **$P = 0.0021$; *$P = 0.0473$; $P = 0.0693$, ns = not significant. **G** Representative H&E staining images of trabecular bone in femur. Black arrows point to trabecular bone. $n = 5$ mice/group. Scale bar: 100 µm. **H, I** Representative µCT images of femur trabecular bones and quantitative analyses of trabecular bone volume to total volume (BV/TV) in treated mice. Scale bars: 100µm. $n = 8$ mice (Veh), 7 mice (Doxo), 7 mice (Veh+AP), and 7 mice (Doxo+AP). **$P = 0.0052$; **$P = 0.0099$; $P = 0.9998$, ns = not significant. **J–L** TRAP staining of femur sections to identify osteoclast (arrows) and quantification of the number of osteoclasts per bone surface (N.Oc/BS) and osteoclast surface area per bone surface (Oc.S/BS). Scale bar: 100µm. $n = 10$ mice (Veh), 10 mice (Doxo), 8 mice (Veh+AP), and 10 mice (Doxo+AP). ****$P < 0.0001$; **$P = 0.0038$; $P > 0.9999$, ns = not significant. Data are represented as mean ± SEM and significance was determined by one-way ANOVA with Tukey test. Source data for this figure are provided as a Source Data file.

stained with an anti-CD45.2 and anti-CD45.1 antibody and we found that mice obtained greater than 90% chimerism (Fig. S5B, C). Recipient mice were then treated with PTX ± AP and 9 days later bone density was assessed by µCT. We found that AP-treatment failed to prevent PTX-induced bone loss (Fig. S5D), indicating that p16 positive senescent bone marrow-derived CD45$^+$ cells did not contribute to chemotherapy-induced bone loss.

## Chemotherapy induces senescence in bone marrow adipo-lineage cells

Farr et al. recently demonstrated that mouse osteocytes (OCY) undergo senescence with aging and contribute to bone loss[8]. However, we failed to detect elevated levels of Cdkn2a (p16) gene expression in OCY-enriched bone fractions as evidenced by DMP expression (Fig. S5E) from mice treated with Doxo (Fig. S5F), indicating that chemotherapy does not induce senescence in osteocytes.

To identify which bone resident cells senesced in response to chemotherapy, we performed single-cell RNA-sequencing (scRNA-Seq) on cells isolated from the bone marrow following MACs depletion of CD45$^+$, CD71$^+$, and Ter119$^+$ cells (referred to as the bone resident enriched fraction, Fig. 3A). Our transcriptomic profiling identified a total of six clusters in both Veh- and Doxo-treated mice (Fig. 3B, C): Cxcl12-abundant reticular (CAR) cells, immature osteoclasts, two fibroblast populations, osteoblasts, and chondrocytes, as defined by distinct gene expression patterns (Fig. S5G). Next, we performed a gene set enrichment analyses (GSEA) and found that when we compared the different Doxo-treated cell populations to their corresponding Veh-treated population, the only one that showed an enrichment was the CAR cell population. Indeed, the Doxo-CAR cell population showed an enrichment of the CHICAS_RB1_TARGETS_Senescence and the FRIDMAN senescence signature (Fig. 3D, E). Based on the CHICAS_RB1_TARGETS gene set, we found that only the Doxo-treated CAR cells exhibit a senescence-associated gene signature compared to Veh-treated CAR cells as well as other Doxo- and Veh-treated clusters (Fig. 3F, S5H). Next, to determine which CAR cells expressed p16, we sub-clustered the CAR cells and found 3 subpopulations (Fig. 3G). Among them, only the CAR cell population 1 exhibited elevated Cdkn2a expression levels and reduced levels of the cell proliferation marker (Mki67) in the Doxo group (Fig. 3H). Furthermore, we analyzed a publicly available human bone scRNA-seq dataset (GEO: GSE230295) derived from acute lymphoblastic leukemia patients receiving NOPHO2008, an aggressive chemotherapeutic regimen that includes anthracyclines (e.g. daunorubicin or doxorubicin)[18]. Our transcriptomic profiling identified a total of seven distinct clusters (Fig. S5I): two T cell populations, hematopoietic stem cells (HSCs), CAR cells, B cells, and two erythroid progenitor populations, each defined by characteristic gene expression profiles (Fig. S5J). Gene set enrichment analysis (GSEA) revealed that the CAR cell population was enriched for the FRIDMAN senescence signature (Fig. 3I). Consistent with this, senescence-associated signatures were most prominent in the CAR cell population, which displayed elevated expression of Cdkn2a, Cdkn1a, and Glb1, along with reduced levels of the proliferation marker Mki67 (Fig. 3J) compared to other cell populations. This analysis demonstrates that chemotherapy induces senescence in human CAR cells.

To confirm that p16 was increased in CAR cells, we turned to a senescence reporter mouse. For these studies, we utilized the recently characterized p16-Cre$^{ERT2}$/tdTomato mouse in which the first exon of the endogenous p16$^{INK4a}$ gene was substituted with a Cre$^{ERT2}$ gene, which allowed us to identify p16 expressing cells (tdTom$^+$) at the single-cell level[19]. For these studies, 12-week-old p16-Cre$^{ERT2}$/tdTomato mice were treated with either Doxo or PTX and 9 days later we assessed tdTom$^+$ cells in the femurs. As expected, we found tdTom$^+$ cells present in the marrow space of the femurs and failed to see tdTom$^+$ OCY (Fig. S6A). Analysis of the femurs revealed the increased tdTom$^+$ cells in the metaphysis region of the femur of Doxo and PTX treated mice (Fig. 4A, B, S6B, S6C). To confirm that these p16 positive cells were senescent, we assessed SA-β-gal by using the fluorescent SPiDER-β-gal probe. We found that PTX-treated mice contained increased numbers of double-positive SPiDER$^+$; tdTom$^+$ cells (Fig. S6D), indicating that they were senescent.

To determine if p16 was expressed in CAR cells, we first enriched for CAR cells by sorting CD45$^-$Ter119$^-$CD31$^-$Sca1$^-$Pdgfrβ$^+$ cells using bone tissue including marrow as previously described[20] (Fig. S6E) and carried out RT−qPCR analysis. We found that Doxo treatment increased Cdkn2a expression in CAR cells compared to CAR cells isolated from Veh-treated mice (Fig. 4C). We also noted that the tdTom$^+$ signal was associated with both smaller cells and a large ring-like structure that was reminiscent of mature adipocytes (BMAds) (Fig. S6B). To ask if BMAds, which would not survived the isolation that we performed for our scRNA-seq and thus were absent in the analysis, were senescent, we used an approach that allows you to isolate them without destroying them[21,22]. Using this approach, we isolated BMAds and examined their gene expression by RT-qPCR. As expected BMAds expressed significant levels of Cdkn2a (p16), Cdkn1a (p21), and Il6 following Doxo treatment (Fig. 4D−F) indicative of senescent BMAds (senBMAds). Next, to ask if these cells were important in our model, we treated INK-ATTAC mice with Doxo or Veh ± AP and carried out SA-β-gal/oil red O co-staining and immunostaining. While we observed no difference in p16$^+$/PPARγ$^-$ cells, Doxo treatment increased p16$^+$/PPARγ$^+$ cells that were decreased upon AP treatment in INK-ATTAC mice (Fig. 4G, H, S6F). Further, we found that the femurs from Doxo-treated mice also contained increased numbers of SA-β-gal/oil red O-double positive cells that was again abrogated by AP treatment (Fig. S6G, S6H). Next, confirm that p16 was absent in fibroblasts and chondrocytes after 9 days of Doxo treatment, we used the p16-Cre$^{ERT2}$/tdTomato mouse and stained with antibodies for fibroblast (S100a4) and chondrocyte (Sox9) specific markers. We failed to find colocalization of p16-driven tdTom with S100a4 (Fig. S6I) or Sox9 (Fig. S6J). Additionally, we did not observe p16-driven tdTomato signal on the bone surface, indicating that osteoclasts and osteoblasts did not undergo p16-related senescence (Fig. S6K, L).

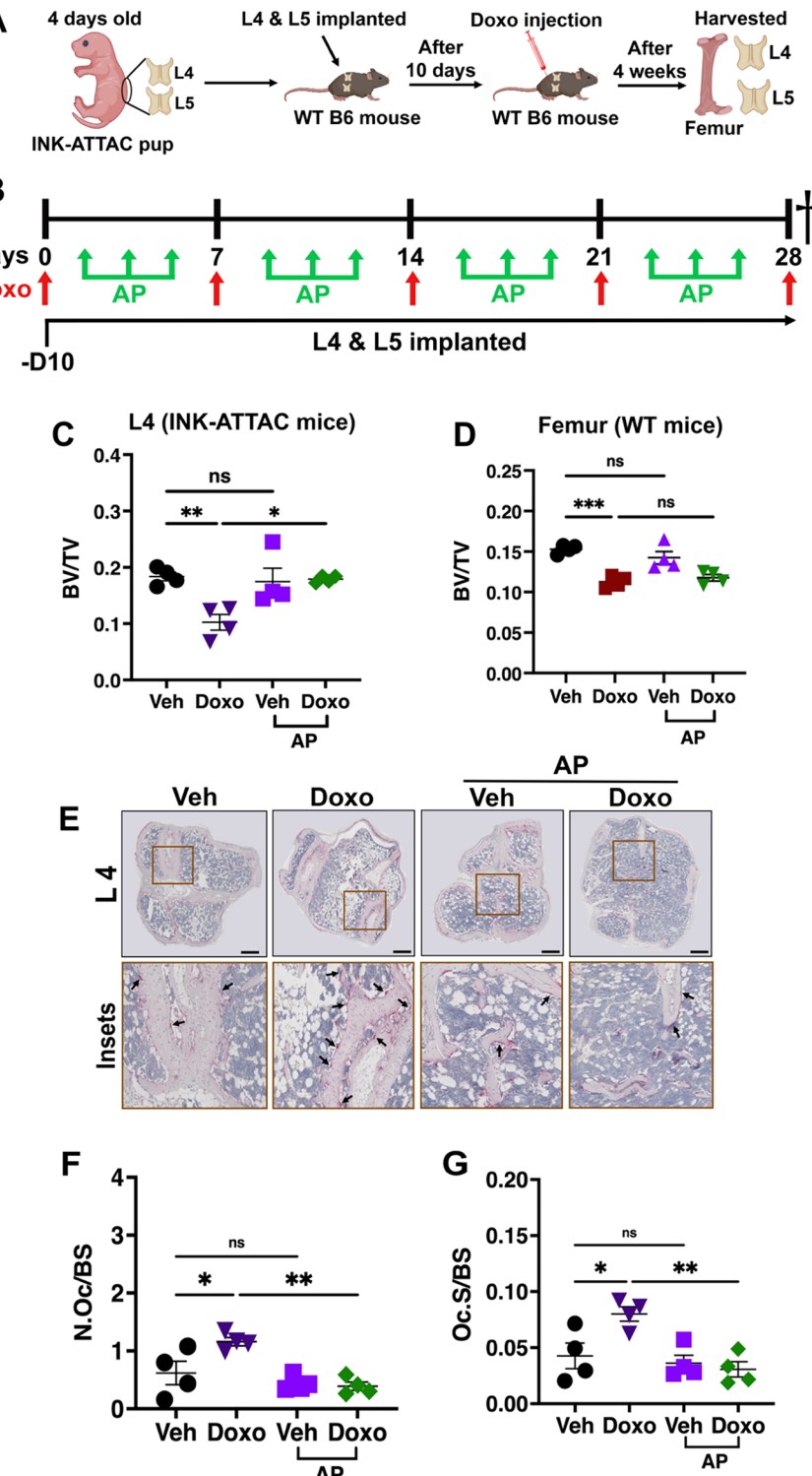

**Fig. 2 | Senescent bone resident cells drive chemotherapy-induced bone loss.**
**A** Schematic diagram illustrating the experimental design for vertebral bodies L4 and L5 implantation. Schematic was created in BioRender. Stewart, S. (2025) https://BioRender.com/71wz45j. **B** Schematic diagram of Doxo and AP treatment timeline. Dagger indicates time of sacrifice. **C, D** μCT quantitative analyses of trabecular bone volume to total volume (BV/TV) for transplanted vertebral bodies L4 (**C**) and WT femur (**D**) as indicated. *n* = 4 mice/group. \*\**P* = 0.0081; \**P* = 0.0122; *P* = 0.9705, ns = not significant (**C**). \*\*\**P* = 0.0004; *P* = 0.8903 (Doxo±AP);

*P* = 0.4602, ns = not significant (**D**). **E**–**G** TRAP staining of femur sections for osteoclasts (arrows) and quantification of the number of osteoclasts per bone surface (N.Oc/BS) and osteoclast surface area per bone surface (Oc.S/BS) for vertebral bodies L4. Scale bar: 100 μm. Insets represents magnified view. *n* = 4 mice/group. \*\**P* = 0.0313; \*\**P* = 0.0029; *P* = 0.7076, ns = not significant (**F**). \*\**P* = 0.0318; \*\**P* = 0.0052; *P* = 0.9400, ns = not significant (**G**). Data are represented as mean ± SEM and significance was determined by one-way ANOVA with Tukey test. Source data for this figure are provided as a Source Data file.

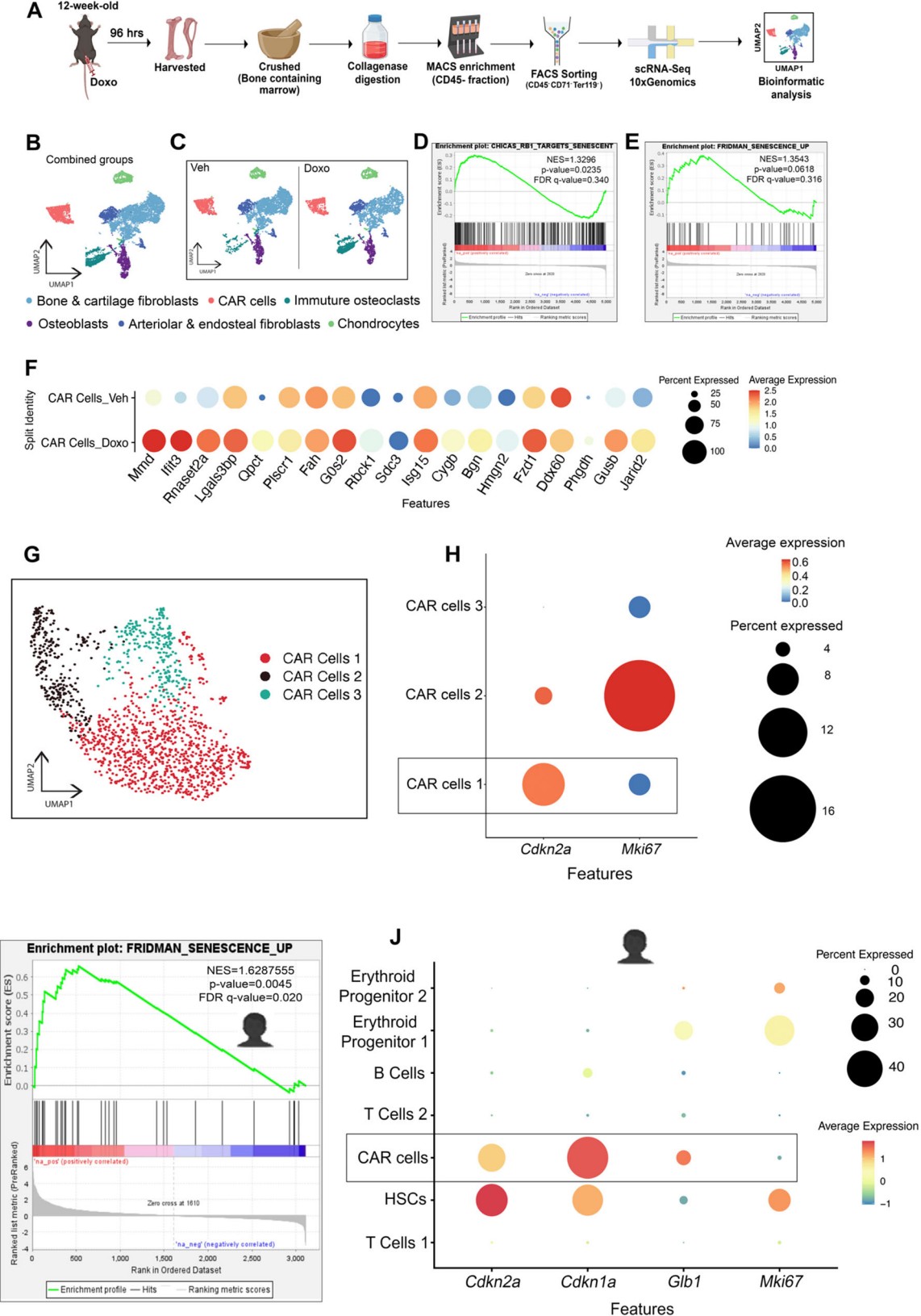

Our scRNA-Seq suggested that CAR cells were senescent and this was supported by our finding that tdTom[+] cells were present in the marrow of Doxo-treated mice (Fig. 4A). To confirm that CAR cells underwent senescence, we used the Cxcl12-driven GFP reporter mouse and stained bone sections for p16 following Doxo treatment. We found that GFP and p16 colocalized in Doxo-treated mouse femurs (Fig. S6M,

N). We also stained femurs from p16-Cre[ERT2]/tdTomato mice for EBF3, a transcription factor preferentially expressed by CAR cells[20]. We found that both Doxo and PTX treatments led to an increase in the number of EBF3[+];tdTom[+] as well as EBF3[−];tdTom[+] cells (Fig. 4I, J, S6O). A similar pattern was observed with PTX treatment (Fig. S6P, S6Q), suggesting that the EBF3[−];tdTom[+] population includes mature adipocytes that

**Fig. 3 | Chemotherapy induces senescence in bone marrow adipo-lineage cells.** **A** Schematic representation illustrating the processing of whole bone (containing marrow) and sorting for single-cell RNA sequencing. Schematic was created in BioRender. Stewart, S. (2025) https://BioRender.com/71wz45j. **B, C** UMAP visualization of single cell transcriptomes displaying six clusters in combined groups (**B**) and in split groups (Veh and Doxo) (**C**). The default assay was set to RNA. **D, E** Gene set enrichment analyses (GSEA) showed CHICAS_RB1_TARGETS and FRIDMAN senescence signature in Doxo-treated CAR cell versus Veh-treated CAR cell. NES, p-value, and FDR q value are shown within the plot. Default assay was set to RNA. The nominal p-value was generated using a permutation test in the GSEA analysis. **F** Dot plot showing the CHICAS_RB1_TARGETS senescence signatures in Doxo- and Veh-treated CAR cell populations. Default assay was set to ALRA. **G** UMAP of three CAR cell populations. Default assay was set to RNA. **H** Dot plot showing the expression of *Cdkn2a* and *Mki67* in CAR cell populations. Default assay was set to ALRA. **I** Gene set enrichment analysis (GSEA) showed FRIDMAN senescence signature in CAR cell populations versus other cell population in human bone samples. NES, p-value, and FDR q value are shown within the plot. Default assay was set to RNA. The nominal p-value was generated using a permutation test in the GSEA analysis. Schematic was created in BioRender. Stewart, S. (2025) https://BioRender.com/71wz45j. **J** Dot plot showing the expression of *Cdkn2a, Cdkn1a, Glb1* and *Mki67* across the cell populations in human bone samples. Schematic was created in BioRender. Stewart, S. (2025) https://BioRender.com/71wz45j. Source data for this figure are provided as a Source Data file.

undergo senescence in response to these chemotherapeutic agents. Finally, to confirm these findings in the INK-ATTAC model we stained for EBF3 and SPiDER (the antibodies for EBF3 and p16 are the same species so we did not use p16 for these stains). We found that Doxo-induced EBF3+;SPiDER+ cells were reduced in INK-ATTAC mice that were also treated with AP (Fig. 4K, L) with no changes in EBF3;SPiDER+ cells in Veh and Doxo groups (Fig. S6R). Together, these data argue that chemotherapy-induced p16+ senescence was restricted to adipo-lineage cells.

### Senescent adiponectin positive cells drive chemotherapy-induced bone loss

Having demonstrated that CAR cells and BMAds senesce in response to chemotherapy, we next asked if they were sufficient to drive chemotherapy-induced bone loss. To test this, we crossed adiponectin-Cre (ADQ-Cre) mice, which targets CAR cells and BMAds[23] to mice bearing the diphtheria toxin receptor (DTR), downstream of a floxed stop codon, creating ADQ-Cre/DTR mice. Diphtheria toxin (DT) injection is non-toxic in wildtype mice and exclusively targets cells expressing the DTR. To characterize the ADQ-Cre/DTR mice, 11-week-old mice were treated with DT (100 ng/mouse/day) for 5 consecutive days (Fig. S7A). DT injection led to efficient ablation of epididymal white adipose tissue (eWAT) and the bone marrow-associated adipocytes (BMAds) (Fig. S7B–D) in ADQ-Cre+ mice, but not in Cre- mice as expected. To assess the role of CAR cells and adipocytes in chemotherapy-induced bone loss, mice received DT followed by a single dose of Doxo (Fig. S7A). Nine days later, we stained bone sections to evaluate p16+ cells. As expected, there was no difference in the numbers of p16+ cells in the Veh- versus Doxo-treated ADQ-Cre+/DTR mice (Fig. S7E and S7F). Subsequently, we performed µCT analysis of the femurs. As reported early, ADQ-Cre+/DTR mice treated with Veh showed a significant increase in femur trabecular bone (BV/TV)[24]. Surprisingly, ADQ-Cre+/DTR mice failed to lose bone following Doxo treatment in both female (Fig. 5A, B, S7G–J) and male mice (Fig. S7K, L) with no changes in cortical bone thickness in female mice (Fig. S7M).

Given that the ADQ-Cre+/DTR mouse displays increased baseline BV/TV and all adipocytes throughout the mouse are lost, we created a mouse model that allowed us to eliminate only p16+ senescent adiponectin-positive cells. To accomplish this, we cloned a lox-stop-lox cassette 5′ of the INK-ATTAC allele followed by a P2 cleavage site and tdTomato (Fig. S8A). This construct was knocked into the ROSA26 safe harbor locus and the resulting QR mouse was mated to the ADQ-Cre mouse. When the ADQ-Cre/QR mouse was treated with Doxo and AP, EBF3+;SPiDER+ cells were eliminated (Fig. S8B, C). µCT analysis demonstrated that targeting these p16+ adiponectin positive cells effectively prevented chemotherapy-induced bone loss (Fig. 5C, D, S8D–G). These data demonstrate that adiponectin positive senCAR cells and senBMAds drive chemotherapy-induced bone loss.

Previously we demonstrated that chemotherapy altered osteoclast and osteoblast numbers (Fig. 1J–L, S3L and S3M). To ask if adipocyte ablated mice restored these numbers, we assessed the osteoclast population in DT-treated ADQ-Cre+/DTR mice and found that osteoclast numbers remained unchanged between the Veh- & Doxo-treated groups (Fig. S8H–J). Similarly, osteoblast numbers were also maintained in response to Doxo (Fig. S8K, L). Taken together, these results demonstrated that BM adipo-lineage cells drive chemotherapy-induced bone loss by altering osteoclast and osteoblast numbers.

Given CAR cells, which are OB and adipocyte progenitors undergo chemotherapy induced senescence, we next questioned if this would result in reduced OB differentiation. To address this, bone marrow cells were isolated from Doxo- and Veh-treated mice nine days after treatment, plated at the same density, and cultured in osteoblastic differentiation medium for up to 21 days as previously described[25]. We found that despite plating the same number of bone marrow cells (1×10^4 cells/per well), cells from Doxo treated mice had reduced osteogenic differentiation potential at day 21 (Fig. S9A). In addition, ALP activity at day 14 (Fig. S9B) and mineralization capacity as assessed by alizarin red S staining at day 21 (Fig. S9C) were dramatically reduced. The reduction in mineralization was consistent with our in vivo findings (Fig. S3N, S3O). While perhaps not surprising given the precursors are sensing in response to chemotherapy, these results demonstrate that osteoblastic differentiation from Doxo-treated bone marrow cells is severely compromised. To ask if adipocyte differentiation was also impacted by Doxo, we cultured bone marrow cells from Doxo-treated mice in adipogenic medium. After 14 days, we found that adipogenic differentiation was reduced (Fig. S9D, E).

### Senescent BM adipo-lineage cells drive bone loss through increase expression of RANKL

Given the significant increases in osteoclasts following chemotherapy treatment, we wanted to ask if senescent CAR cells and adipocytes were responsible for osteoclast differentiation. In the current study we showed that osteocytes (OCY) do not senesce in response to chemotherapy, however they can regulate osteoclastogenesis through RANKL (Tnfsf11) secretion[26]. To ask if senescent CAR cells/BMAds drove osteocytes to express, we evaluated the *Rankl/Opg* (OPG, a negative regulator of osteoclastogenesis) gene expression ratio in the OCY-enriched bone fraction using RT-qPCR. We found no significant differences in the *Rankl/Opg ratio* between Doxo- and Veh-treated groups in the OCY-enriched bone fraction (Fig. S10A–C). While we failed to observe changes in OCYs, gene expression analysis of the bone-resident fraction from 12-week-old C57LB/6 mice treated with PTX showed a high *Rankl/Opg ratio* compared to Veh-treated samples (Fig. S10D–F), raising the possibility that senescent CAR and/or BMAd were the source of RANKL.

To ask if the senescent CAR/BMAds impacted the *Rankl/Opg ratio*, we returned to the INK-ATTAC transgenic mouse model (Fig. 6A). Immunohistochemistry staining and RT-qPCR results showed that the elimination of senescent CAR/BMAds by AP-treatment abolished Doxo-induced *Rankl* production and reduced the *Rankl/Opg ratio* in the bone-resident fraction (Fig. 6B–F). Accumulating evidence has revealed that marrow adipo-lineage cells including CAR cells and BMAds can express RANKL in pathological conditions of bone loss[1,27,28],

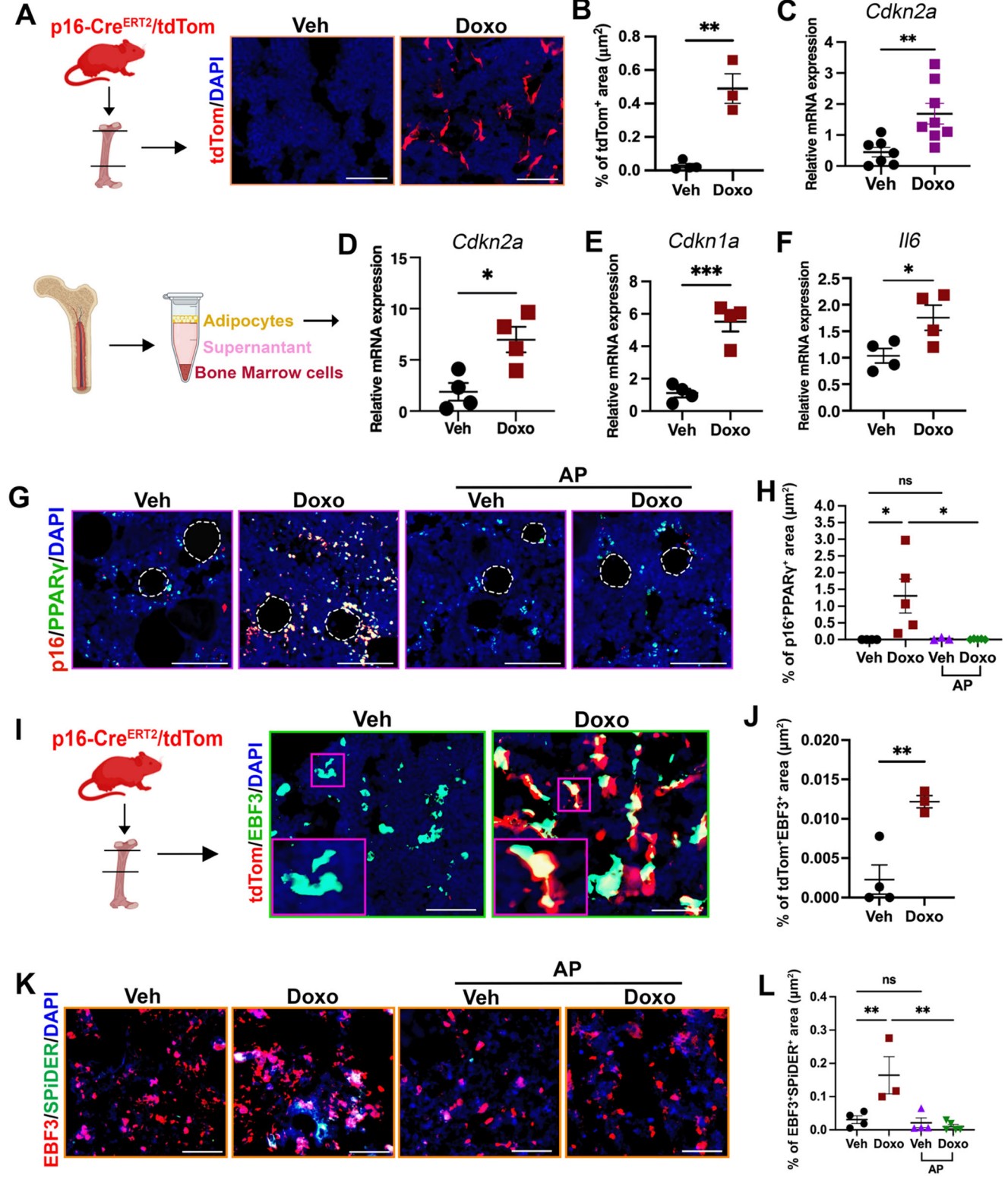

and our scRNA-Seq analysis suggested elevated expression levels of *Rankl* in Doxo-treated CAR cells compared to Veh-treated CAR cells (Fig. S10G), suggesting CAR cells were responsible for osteoclast differentiation. To confirm that senCARs and senBMAds were the source of increased RANKL in response to chemotherapy, we isolated floating senBMAds and sorted to enrich for senCARs (CD45⁻Ter119⁻CD31⁻Sca1⁻Pdgfrβ⁺) as previously described and found that the *Rankl/Opg ratio* was increased in cells from Doxo-treated versus Veh-treated mice (Fig. 6G–M), indicating that chemotherapy-

induced senCARs and senBMAds support osteoclastogenesis through RANKL secretion.

To establish that it was BM adipo-lineage cell-derived RANKL that drove bone loss in response to chemotherapy, we first performed gene expression analysis of the *Rankl/Opg ratio* using the bone-resident fraction from 11-week-old ADQ-Cre+/DTR mice, we found no differences in cells from the Doxo- and Veh-treated groups (Fig. S10H–J). Thus, we next crossed the ADQ-Cre^ERT2 mouse to Tnfsf11^fl/fl mice to allow us to delete *Rankl* conditionally (referred to as ADQ-Cre/Rankl)

**Fig. 4 | Chemotherapy induces senescence in bone marrow adipo-lineage cells.**
**A** Representative images of tdTomato-expressing cells in the distal femur of 12-week-old p16-Cre^ERT2/tdTomato mice treated with Veh or Doxo. DAPI stained nuclei blue. Scale bar: 100µm. Schematic was created in BioRender. Stewart, S. (2025) https://BioRender.com/71wz45j. **B** Percentages of tdTom+ cells area in femur sections of p16-Cre^ERT2/tdTomato mice. *n* = 3 mice (Veh) and 4 mice (Doxo). **P* = 0.0017. **C** RT-qPCR analyses of p16^INK4a expression in CD45−Ter119−CD31−Sca1−Pdgfrβ+ sorted cells enriched for CAR cells in Veh or Doxo-treated mice. Actin and cyclophilin were used as housekeeping genes. *n* = 7 mice (Veh) and 8 mice (Doxo). **P* = 0.0068. **D−F** RT-qPCR analyses of *Cdkn2a, p21* and *Il6* in isolated adipocytes in indicated treatment. Actin and cyclophilin were used as housekeeping genes. *n* = 4/group. *P* = 0.0152 (**D**), ***P* = 0.0005 (**E**), *P* = 0.0398 (**F**). **G, H** Co-immunofluorescence staining showing colocalization of p16 (red) and PPARγ (green) in femoral bone sections of INK-ATTAC mice. DAPI stained nuclei are in blue. Dotted lines outline adipocytes. Quantification of the percentage of p16 and PPARγ-double positive cells area. Scale bar: 50 µm. *n* = 4 mice (Veh), 5 mice (Doxo), 3 mice (Veh+AP), and 5 mice (Doxo+AP). *P* = 0.0381 (Veh *vs* Doxo); *P* = 0.0286 (Doxo±AP); *P* > 0.9999, ns = not significant. **I, J** Representative images showing tdTom expression (red) co-localized with EBF3 (green) in the femurs of p16-Cre^ERT2/tdTomato mice. DAPI stained nuclei are blue. Scale bar: 50 µm. *n* = 4 mice (Veh) and 3 mice (Doxo). **P* = 0.0313. Schematic was created in BioRender. Stewart, S. (2025) https://BioRender.com/71wz45j. **K, L** Immunofluorescence staining showing colocalization of EBF3 (red) and SPiDER (green) in femoral bone sections of INK-ATTAC mice. DAPI stained nuclei are blue. Quantification of the percentage of EBF3 and SPiDER double positive cell area. Scale bar: 50µm. *n* = 4 mice (Veh), 3 mice (Doxo), 4 mice (Veh+AP), and 5 mice (Doxo+AP). **P* < 0.0094 (Veh *vs* Doxo); **P* = 0.0024 (Doxo±AP); *P* = 0.9895, ns = not significant. Data are represented as mean ± SEM and significance was determined by unpaired two-tailed Student's *t*-tests and one-way ANOVA with Tukey test. Source data for this figure are provided as a Source Data file.

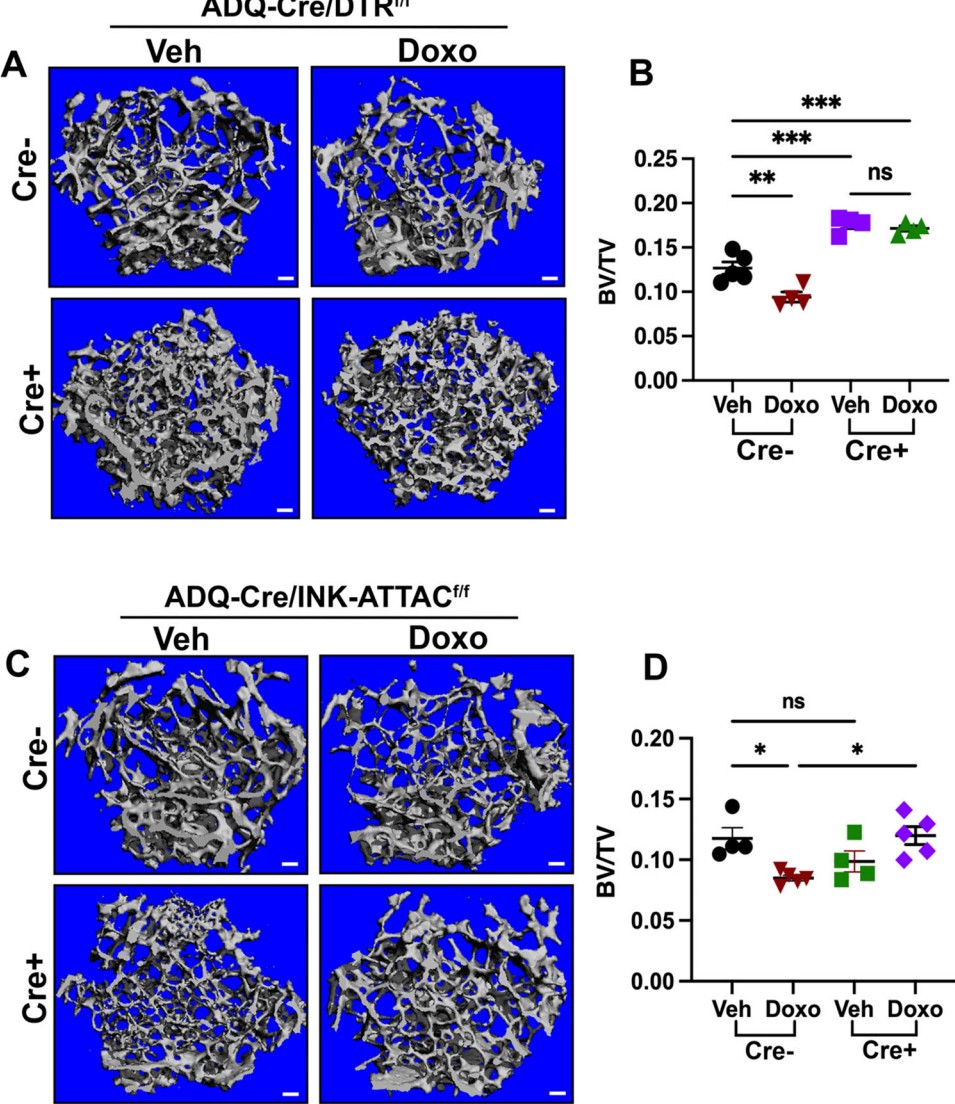

**Fig. 5 | Ablation of bone marrow adipo-lineage cells prevent chemotherapy-induced bone loss. A, B** Representative µCT images and quantitative analysis of femurs from ADQ-Cre/DTR^f/f mice. Scale bar: 100µm. *n* = 5 mice (Veh/Cre-), 4 mice (Doxo/Cre-), and 4 mice (Cre+). **P* < 0.0050 (Veh *vs* Doxo); ***P* = 0.0001; ***P* = 0.0003 (Veh/Cre- *vs* Doxo/Cre+). *P* = 0.9441, ns = not significant. **C, D** Representative µCT images and quantitative analysis of femurs from ADQ-Cre/INK-ATTAC^f/f (QR-mice). Scale bar: 100µm. *n* = 4 mice (Veh), 5 mice (Doxo), 4 mice (Veh +AP), and 5 mice (Doxo+AP). *P* = 0.0238; *P* = 0.0102 (Doxo±AP). *P* = 0.3066, ns = not significant. Data are represented as mean ± SEM and significance was determined by one-way ANOVA with Tukey test. Source data for this figure are provided as a Source Data file.

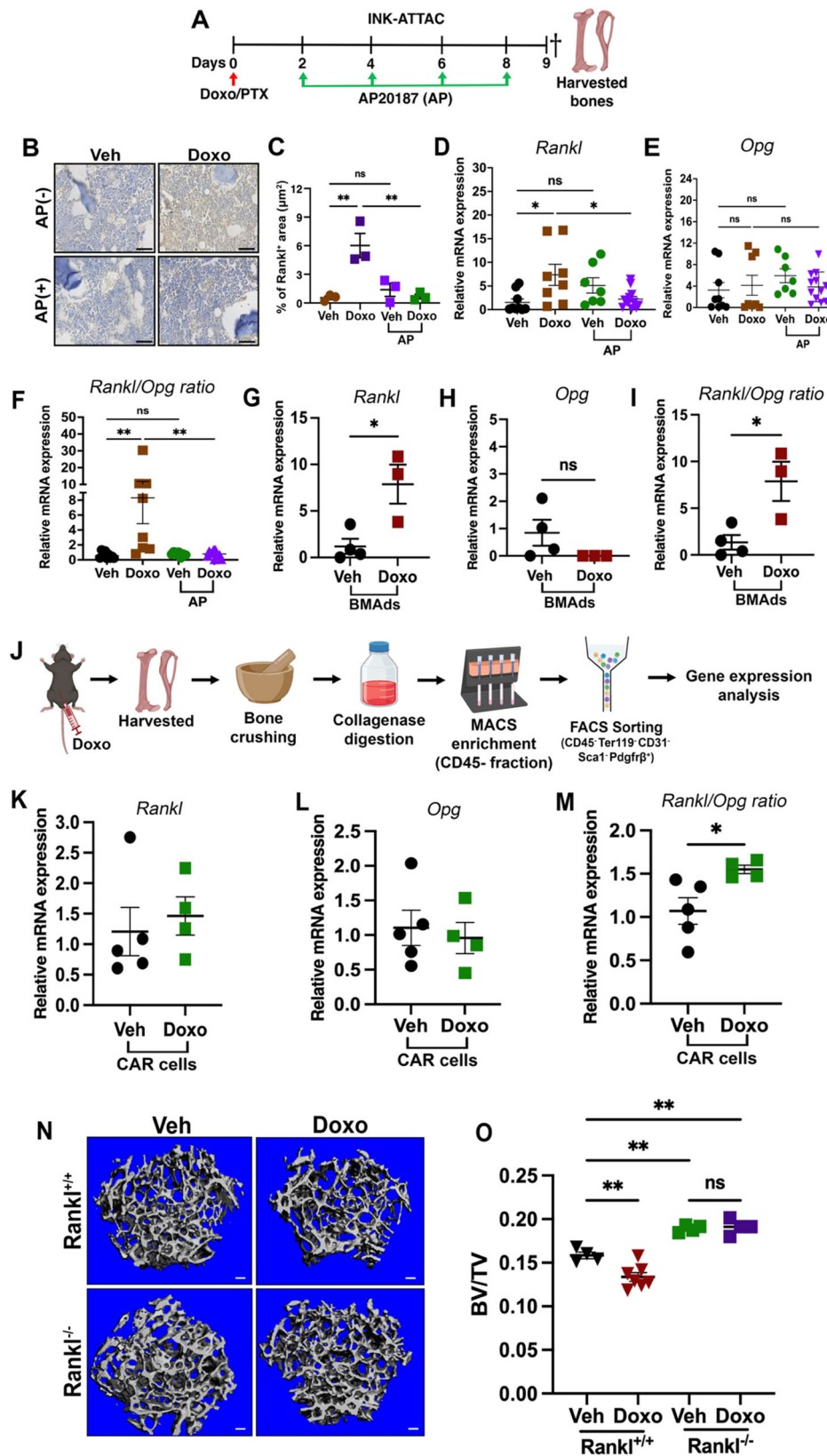

in BM adipose lineage cells (Fig. S10K). Prior to using these mice, we confirmed that *Rankl* was deleted upon Cre activation in femurs (Fig. S10L). To investigate whether BM adipo-lineage cell derived RANKL contributes to osteoclastogenesis in response to chemotherapy, we carried out TRAP staining on ADQ-Cre/Rankl mice and found reduced numbers of osteoclasts in Doxo-treated Rankl$^{-/-}$ mice

compared with Doxo-treated control mice (Fig. S10M−O). Given this reduction in osteoclasts, we next asked how BV/TV was impacted in these mice. Thus, we harvested the femurs from Veh- and Doxo-treated 12-week-old female ADQ-Cre/Tnfsf11$^{+/+}$ mice. Importantly, immunostaining revealed that p16$^+$ cells remained detectable in *Rankl*-deficient mice, indicating that *Rankl* deletion does not prevent the

**Fig. 6 | Senescent bone marrow adipo-lineage cells drive bone loss through increased expression of RANKL. A** Schematic showing experimental timeline for dosing regimen for doxorubicin (Doxo) and paclitaxel (PTX) in 12-week-old INK-ATTAC mice. Dagger indicates time of sacrifice and bone harvest. Schematic was created in BioRender. Stewart, S. (2025) https://BioRender.com/71wz45j. **B, C** Representative IHC staining for the detection of RANKL expression in INK-ATTAC mice femur under the indicated treatments. Scale bar: 100μm. $n = 3$ mice/group. **$P = 0.0035$; **$P = 0.0041$ (Doxo±AP), $P = 0.8525$, ns = not significant. **D–F** mRNA expression of *Rankl, Opg and Rankl/Opg ratio* as quantified by RT-qPCR using the bone-resident fraction of INK-ATTAC mice. $n = 9$ mice (Veh), 8 mice (Doxo), 7 mice (Veh+AP), and 12 mice (Doxo+AP). Actin and cyclophilin were used as housekeeping genes. *$P = 0.0186$; *$P = 0.0295$ (Doxo±AP) (**D**), ns = not significant (**E**), **$P = 0.0093$; **$P = 0.0081$ (Doxo±AP) (**F**). **G–I** mRNA expression of *Rankl, Opg and Rankl/Opg ratio* as quantified by RT-qPCR in purified bone marrow adipocytes. Actin and cyclophilin were used as housekeeping genes. $n = 4$ mice (Veh) and 3 mice

(Doxo). *$P = 0.0206$ (**G**). $P = 0.0206$ (**H**). *$P = 0.0216$ (**I**), ns = not significant. **J** Schematic represents CAR cell sorting procedure from mouse bone. Schematic was created in BioRender. Stewart, S. (2025) https://BioRender.com/71wz45j. **K–M** mRNA expression of *Rankl, Opg* and *Rankl/Opg ratio* as quantified by RT-qPCR in CD45⁻Ter119⁻CD31⁻Sca1⁻Pdgfrβ⁺ sorted CAR cells. Actin and cyclophilin were used as housekeeping genes. $n = 5$ mice (Veh) and 4 mice (Doxo). *$P = 0.0311$. **N, O** μCT 3D trabecular bone images and quantitative analyses of BV/TV in 12-week-old female adipo-Cre-/Rankl^fl/fl and ADQ-Cre+/Rankl^fl/fl mice. $n = 4$ mice (Veh/Rankl⁺/⁺), 7 mice (Doxo/Rankl⁺/⁺), 4 mice (Veh/Rankl⁻/⁻), and 4 mice (Doxo/Rankl⁻/⁻). **$P = 0.0060$ (Rankl⁺/⁺:Veh/Doxo); **$P = 0.0027$ (Rankl⁺/⁺:Veh vs Rankl⁻/⁻:Veh); **$P = 0.0015$ (Rankl⁺/⁺:Veh vs Rankl⁻/⁻:Doxo), ns = not significant. Data are represented as mean ± SEM and significance was determined by unpaired two-tailed Student's *t*-tests and one-way ANOVA with Tukey test. Source data for this figure are provided as a Source Data file.

accumulation of senescent cells (Fig. S10P, Q). μCT analyses revealed that deletion of *Rankl* from adiponectin positive cells prevented Doxo-induced bone loss as confirmed by unchanged trabecular bone volume (BV/TV), bone mineral density (BMD), trabecular number (Tb.N), and trabecular spacing (Tb.Sp) in Doxo-treated ADQ-Cre+/Rankl⁺/⁺ mice compared to Veh-treated mice (Fig. 6N, O, S11A–D). We also analyzed femurs from male mice and found the same results (Fig. S11E–I). Collectively, these findings indicate that RANKL derived from senCARs and senBMAds contributes to bone loss in both female and male mice.

## Senolytics and senomorphics prevent chemotherapy-induced bone loss

Our use of the INK-ATTAC mouse model demonstrated that targeting senescent bone cells prevented chemotherapy-induced bone loss. To complement this work and explore the feasibility of pharmacologically targeting senescent bone resident cells in a therapeutic setting, we used a senolytic approach (5 mg/kg dasatinib and 50 mg/kg quercetin cocktail, D + Q) or senomorphics that target the p38MAPKα or MK2 kinase to target chemotherapy-induced senescent cells and SASP, respectively. To establish the specificity of the D + Q approach, we first examined the senescence profile in femurs harvested from 12-week-old female Veh- and Doxo- or PTX ± (D + Q)-treated groups (Fig. 7A). As expected, we observed that D + Q significantly reduced Doxo- and PTX-induced SA-β-gal positive cells (Fig. S12A, B) as well as Doxo-induced EBF3⁺;SPiDER⁺ CAR cells (Fig. 7B, C). Next, to examine whether the elimination of senescent cells by D + Q could prevent chemotherapy-induced bone loss, we performed μCT and found that Doxo-induced bone loss was prevented by D + Q treatment (Fig. 7D, E). Other μCT bone parameters such as BMD, Tb. N, Tb.Th and Tb.Sp were also evaluated and altered as expected (Fig. S12C–F). In addition, the impact of PTX treatment was also tested and we found that D + Q also prevented PTX-induced bone loss (Fig. S12G–K). Furthermore, quantification of osteoclast number by TRAP staining showed a significant increase in osteoclast numbers (N.Oc/BS and Oc.S/BS) in Doxo-treated mice as expected that was prevented by D + Q treatment (Fig. 7F–H). Moreover, Doxo-induced reductions in OCN⁺ osteoblasts were also prevented by D + Q treatment (Fig. S12L, M), indicating that the pharmacologic elimination of senescent cells prevents the disruption of bone homeostasis.

Senescent cells produce a complex mixture of factors collectively termed the senescence-associated secretory phenotype (SASP) that is unique to the cell type undergoing senescence and the stress responsible for the induction of senescence[29,30]. The SASP is comprised of cytokines, chemokines, extracellular matrix proteases, growth factors, and other signaling molecules[30,31]. These factors can alter the tissue environment by activating cell surface receptors and their related signaling pathways, leading to several pathologies, including bone loss[29,32].

Previously, we showed that inhibition of the p38MAPK-MK2 pathway rescued chemotherapy-induced bone loss[6]. Here, we wanted to ask if bone resident senescent cells were impacted by p38MAPKα-MK2 inhibitors (p38i/MK2i). To address this, we performed immuno-fluorescence staining using a p16 antibody and found, as expected, that inhibition of p38MAPKα or MK2 (p38i or MK2i) failed to eliminate senescent cells in the bone (Fig. 7I, J). However, we found that within the bone marrow, the osteoclastogenic cytokine (RANKL), a component of the SASP[33], was significantly reduced by both p38i and MK2i treatments (Fig. S13A, B). Moreover, RT-qPCR gene expression analysis of the bone-resident fraction showed decreased expression of additional SASP factors (e.g., *Il6, MMP9,* and *Vegfα)* in the Doxo+p38i/MK2i group compared to the Doxo-treated group (Fig. S13C), indicating that targeting the p38MAPKα-MK2 pathway can limit senescent cell-derived SASP factors. Subsequently, we asked if we prevented acute bone loss with p38i/MK2i. μCT analysis showed that inhibition of SASP by p38i/MK2i prevented chemotherapy (Doxo and PTX)-induced trabecular bone loss while no change in the cortical bones was observed (Fig. 7K, L, S13D–I, S13J–P). Collectively, these data demonstrate that pharmacological inhibition of the SASP prevents chemotherapy induced bone loss, raising the possibility that this could be a viable approach in the clinical setting.

Some studies have shown that senescent bone marrow stromal cells support osteoclast differentiation by secreting SASP factors such as RANKL that are closely linked to bone loss[8,34,35]. Therefore, next we evaluated osteoclast numbers by TRAP staining in 12-week-old female mice treated with Veh and Doxo±p38i/MK2i. Results showed that Doxo significantly increases osteoclast numbers as expected that were reduced by p38i/MK2i (Fig. 7M–O). We also evaluated osteoblast-mediated bone formation rate over bone surface (BFR/BS) by calcein/alizarin red staining. As expected, bone formation was reduced in Doxo-treated compared to Veh-treated mice after 9 days while Doxo+p38i/MK2i groups restored bone formation activity similar to Veh-treated mice (Fig. 7P, Q). Taken together, these data demonstrate that chemotherapy-induced senescent BM adipo-lineage cells secrete SASP factors that in turn drive bone loss by directly altering the balanced action of osteoclasts and osteoblasts.

The use of the p38/MK2 inhibitors could affect all cells. Thus, we next wanted to confirm whether SASP derived from adiponectin-positive cells was sufficient to drive bone loss in response to chemotherapy. Thus, we mated the ADQ-Cre^ERT2 mouse to a mouse that contained a floxed Mapkapk2 allele (referred to as MK2^f/f herein) to specifically assess the role of MK2 in adiponectin expressing cells. ADQ-Cre^ERT2/MK2^f/f mice were treated with tamoxifen chow for 9 days and received Doxo or vehicle and bones were assessed nine days later (Fig. S14A). To assess the presence of senescent cells, we first stained bones for p16 expression and found that p16⁺ cells were high in the Doxo-treated group (Fig. S14B, C), indicating that loss of MK2 does not

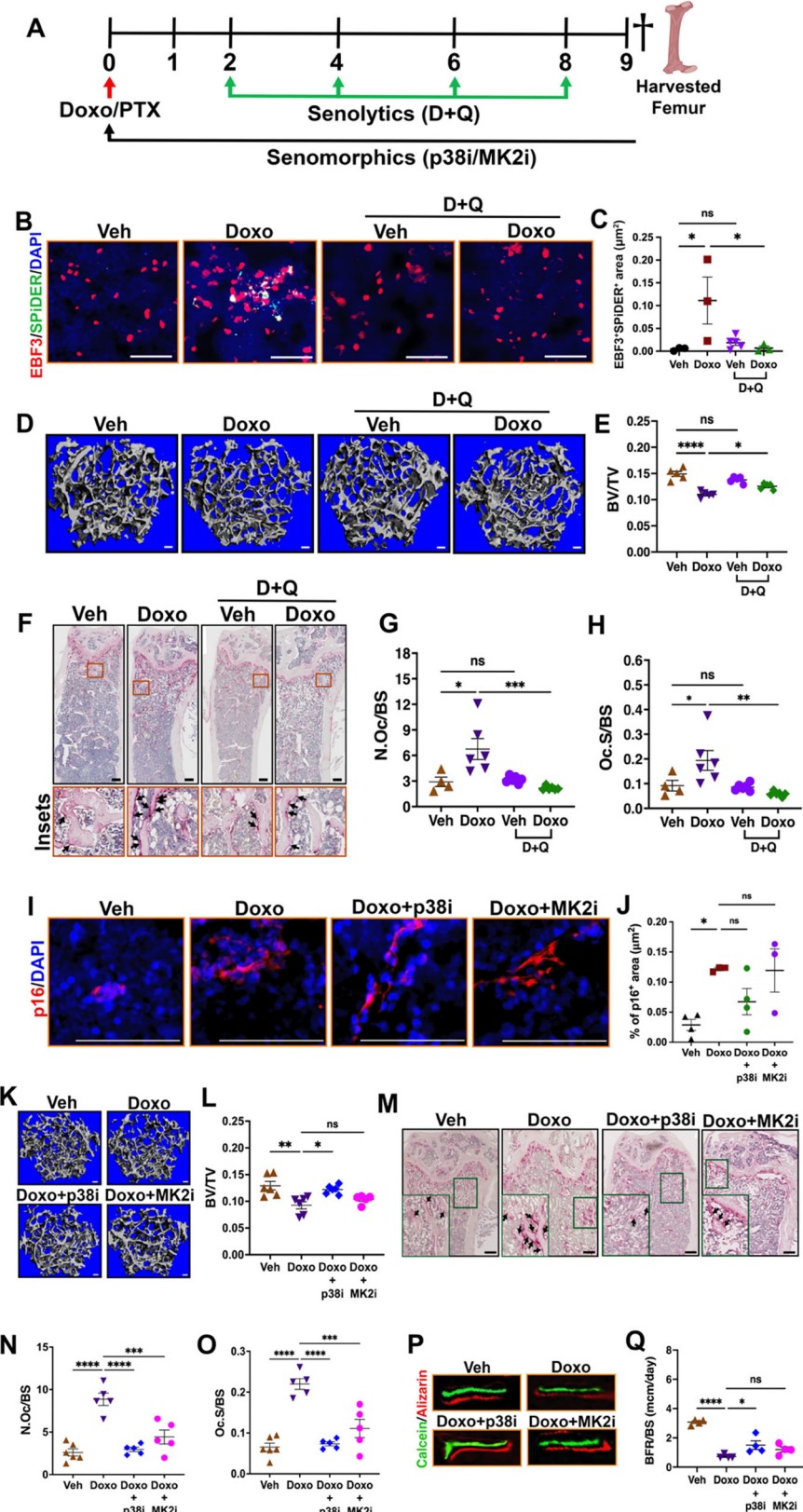

inhibit the induction of senescence. In contrast, when we assessed the bone, we found no significant differences in BV/TV between Veh- and Doxo-treated groups (Fig. S14D), indicating that MK2-dependent SASP derived from adiponectin positive cells drives acute bone loss in response to chemotherapy.

## Senolytics prevent chemotherapy-induced bone loss in tumor bearing mice

Bone loss is a significant problem for breast cancer patients undergoing chemotherapy treatment and can lead to increased fracture risks and reduced overall survival[36,37]. Given the potent effects of senolytics

**Fig. 7 | Senolytics and senomorphics prevent chemotherapy-induced bone loss.** **A** Schematic of the experimental timeline for the injection of chemotherapy (Doxo or PTX), senolytics (D + Q) or senomorphics in 12-week-old C57BL/6 WT-mice. Dagger indicates time of sacrifice and bone harvest. Schematic was created in BioRender. Stewart, S. (2025) https://BioRender.com/71wz45j. **B, C** Representative images of co-localization of EBF3 and SPiDER staining and quantification of EBF3 and SPiDER positive cell area in femurs. Scale bar: 50μm. $n = 3$ mice (Veh), 3 mice (Doxo), 5 mice (Veh+D/Q), and 3 mice (Doxo+D/Q). *$P = 0.0410$; *$P = 0.0456$ (Doxo ±D/Q), ns = not significant. **D, E** Representative 3D μCT images of femur trabecular bones and quantitative analyses of trabecular bone volume to total volume (BV/TV). Scale bar:100μm. $n = 5$/group. ****$P < 0.0001$; *$P = 0.0217$ (Doxo±D/Q), ns = not significant. **F–H** TRAP staining of femur sections showing osteoclasts (arrows). Insets show magnified figures and arrows indicate pink colored osteoclasts. Quantification of the number of osteoclasts per bone surface (N.Oc/BS) and osteoclast surface area per bone surface (Oc.S/BS). Scale bar: 100μm. $n = 4$ mice (Veh), 5 mice (Doxo), 6 mice (Veh+D/Q), and 6 mice (Doxo+D/Q). *$P = 0.0104$; ***$P = 0.0008$, ns = not significant. (**G**). *$P = 0.0467$; **$P = 0.0025$, ns = not significant (**H**). **I, J** Immunofluorescence staining showing p16 (red) positive cells in femoral bone sections of 12-week-old wild type C57BL/6 mice. DAPI stained nuclei are blue. Quantification of the percentage of p16 positive area. Scale bar: 50μm. $n = 4$ mice (Veh), 3 mice (Doxo), 4 mice (Doxo+p38i), and 3 mice (Doxo+MK2i). *$P = 0.0406$; $P = 0.3021$ (Doxo+p38i); $P = 0.9999$ (Doxo+MK2i), ns = not significant. **K, L** Representative μCT images of femur trabecular bones and quantitative analysis of trabecular bone volume to total volume (BV/TV). Scale bar: 100μm. $n = 6$ mice (Veh), 6 mice (Doxo), 6 mice (Doxo+p38i), and 5 mice (Doxo+MK2i). **$P = 0.0015$; *$P = 0.0102$ (Doxo+p38i); $P = 0.5804$ (Doxo+MK2i), ns = not significant. **M–O** TRAP staining of femur sections. Insets show magnified figures and arrows indicate pink colored osteoclasts. Quantification of the number of osteoclasts per bone surface (N.Oc/BS) and osteoclast surface area per bone surface (Oc.S/BS). Scale bar: 100μm. $n = 6$ mice (Veh), 5 mice (Doxo), 5 mice (Doxo+p38i), and 5 mice (Doxo+MK2i). ****$P < 0.0001$; ****$P < 0.0001$ (Doxo+p38i); ***$P = 0.0003$ (Doxo+MK2i) (**N**). ****$P < 0.0001$; ****$P < 0.0001$ (Doxo+p38i); ***$P = 0.0002$ (Doxo+MK2i) (**O**). **P, Q** Representative images of calcein and alizarin red double bone labeling in femurs and analysis of the percentage of double-labeled area over tissue area underneath the growth plate. $n = 4$ mice/group. ****$P < 0.0001$; *$P < 0.0431$ (Doxo+p38i); $P = 0.2888$ (Doxo+MK2i), ns = not significant. Data are represented as mean ± SEM and significance was determined by one-way ANOVA with Tukey test. Source data for this figure are provided as a Source Data file.

on preventing chemotherapy-induced bone loss, we next wanted to ask how senolytics impacted the anti-tumor effects of chemotherapy. Using a clinically relevant mouse model, we asked whether targeting senescent cells by senolytics (D + Q) could rescue chemotherapy-induced bone loss within bones harboring metastatic breast cancer. For these studies, we utilized paclitaxel (PTX) because it is given as part of the neoadjuvant (or adjuvant) chemotherapy regimen for all breast cancer subtypes when chemotherapy is indicated. We injected $5 \times 10^4$ PyMT-BO-1 (GFP/Luc) cells intracardially into 12-week-old female albino C57BL/6 J mice to broadly distribute metastatic cells. Mice were then randomized to treatment groups that consisted of Veh, PTX (50 mg/kg), or PTX + (D + Q). Mice were sacrificed on day 13 when Veh treated mice became moribund (Fig. 8A). We first assessed senescence induction in this model by staining femurs with the SPiDER-β-gal probe and found that D + Q treatment decreased PTX-induced SA-β-gal⁺ cells (Fig. 8B, C). Co-staining for EBF3 and SPiDER revealed D + Q reduced senCARs cells (EBF3⁺;SPiDER⁺) in mouse femurs (Fig. 8D, E). Importantly, bioluminescence imaging of metastatic tumor cells revealed that D + Q treatment did not impact the anti-tumor effects of PTX (Fig. 8F, G). Consistently, H&E staining of tumor-bearing bone tissues further confirmed these findings (Fig. S15A). However, while μCT analysis of the femurs showed significant bone loss in PTX-treated mice, the addition of D + Q prevented trabecular bone loss (Fig. 8H, I) without affecting cortical thickness (S15B, C). Osteoclast numbers were greatly reduced when PTX treated mice also received D + Q (Fig. 8J–L). However, PTX plus D + Q did not lower tumor burden compared to PTX alone in TRAP-stained samples, consistent with BLI and H&E staining results (Fig. 8J, S15D). Whereas osteoblasts (OCN⁺ cells) were significantly increased when mice were treated with PTX and D + Q compared to PTX alone (Fig. S15E, F). Together, these results demonstrate that targeting senCARs and senBMAds to prevent chemotherapy-induced bone loss during cancer treatment preserves the anti-tumor effects of chemotherapy.

## Discussion

Previously we showed that chemotherapy-induced bone loss was associated with senescence but the cell(s) and mechanism(s) responsible remained elusive[6]. In the present study, we show that adiponectin positive cells including CAR cells and BMAds senesce and express SASP factors including RANKL in response to chemotherapy. RANKL expression exclusively from these cells drives osteoclastogenesis and bone loss. Further, senCAR cells and senBMAds simultaneously reduce osteoblast numbers and mineralization capacity that together with the increased osteoclasts drives acute bone loss. Finally, we show that targeting MK2 specifically in adiponectin positive cells is sufficient to

prevent chemotherapy-induced bone loss and that the use of senomorphics (p38i/MK2i) or senolytics (D + Q) can preserve bone homeostasis without negatively impacting the anti-tumor properties of chemotherapy. This latter finding raises the possibility that our approach could drastically improve a patient's quality of life and allow patients to avoid de-escalation of treatment when bone loss becomes extensive, which negatively impacts overall survival.

Senescence has generally been thought of as a systemic and/or local response to stress and thus we had expected chemotherapy to induce senescence in multiple cell types within the bone. Thus, we were surprised to find that p16⁺ senescence was restricted to CAR cells and BMAds. Why senescence was restricted to these two cell types remains an important question. We also found that senCARs and senBMAds expressed SASP factors including RANKL, leading to increased bone resorption. Interestingly, a recent study showed that baseline bone mass is higher in ADQ-Cre/Rankl^{fl/fl} mice[38] supporting the notion that RANKL derived from adiponectin-positive cells plays a critical role not only in pathological bone loss but also in maintaining normal bone homeostasis.

Recent work has demonstrated that the SASP is highly heterogeneous, varies depending on the cell type and the senescence-inducing stimulus, and is dynamic, changing over time[39]. This is underscored by our findings and those by Khosla and colleagues who showed that in the aging bone, senescence is restricted to osteocytes and senescent osteocytes drive age-related bone loss. This contrasts to our findings where chemotherapy fails to induce senescence in osteocytes but instead induces robust senescence in CAR cells/BMAds, which drive bone loss. Why these different inducers, aging and chemotherapy impact different cells remains and important question. Another outstanding question is why senescent CAR cells/BMAds express RANKL and other factors. Finally, bone resorption and formation are tightly coupled and increased osteoclastogenesis is typically accompanied by increased osteoblastogenesis[40]. The induction of senescence in CAR cells/BMAds decouples this association where we find that increased osteoclastogenesis is associated with decreased osteoblastogensis, leading to significant bone loss within nine days of treatment. Future work will need to focus on how senCAR/senBMAd drive reduced osteoblastogenesis.

Currently, >650,000 cancer patients receive chemotherapy every year in the United States[41,42]. Unfortunately, while chemotherapy can increase a cancer patient's overall survival, it also induces a large cadre of comorbidities that significantly impact a patient's quality of life. One such comorbidity is therapy-induced bone loss[36] that can exceed 7% a year in women with breast cancer, which can render a patient susceptible to bone fractures[43]. Currently, the mainstay drugs for patients suffering from chemotherapy-induced bone loss are anti-resorptive

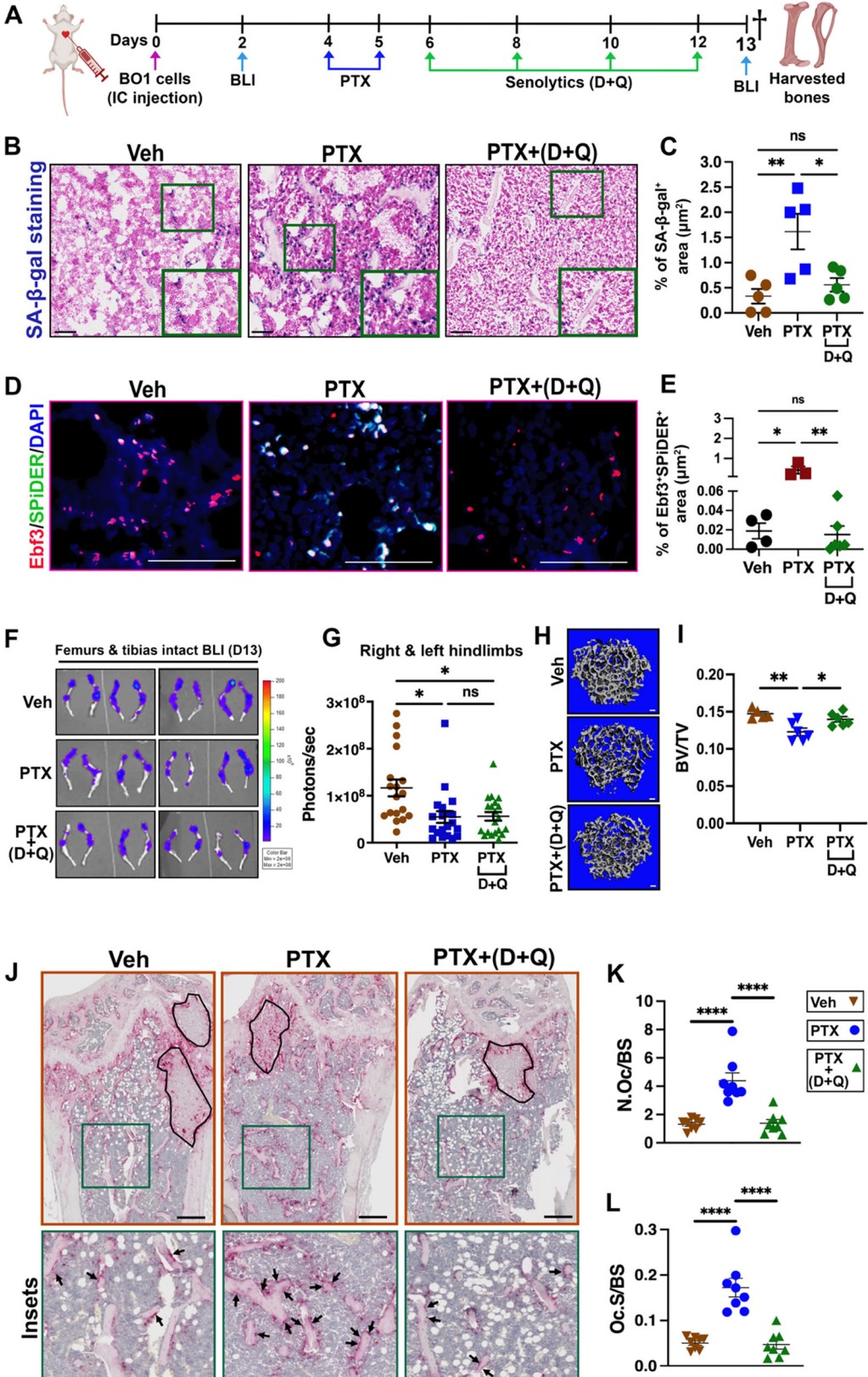

medicines, such as denosumab (anti-RANKL antibody), bisphosphonates[44], and hormone replacement[45]. Unfortunately, hormone replacement is not an option for most breast cancer patients, while bisphosphonates can persist in the bone for long periods and can cause complications such as atypical fractures and osteonecrosis of the jaw[46,47]. Similarly, denosumab has side effects including a

"rebound" when discontinued that can result in significant bone loss, musculoskeletal pain, hypercholesterolemia, and cystitis[48]. Together these data argue that safe and potent therapeutic alternatives to treat chemotherapy-induced bone loss are needed.

Senescent cells are resistant to apoptosis due in part to high expression of Bcl-2 or Bcl-2 family members[49,50]. Despite this,

**Fig. 8 | Senolytics prevent chemotherapy-induced bone loss in tumor bearing mice. A** Schematic of the experimental timeline for the injection of Luciferase[+] BO-1 breast cancer cells and other dosing regimen in C57BL/6 -albino mice. Dagger indicates time of sacrifice and bone harvest. Schematic was created in BioRender. Stewart, S. (2025) https://BioRender.com/71wz45j. **B, C** Representative images showing SA-β-gal staining and quantification of SA-β-gal[+] area. Scale bar: 100 μm. $n = 5$ mice/group. **$P = 0.0058$; *$P = 0.0197$; $P = 0.7764$, ns = not significant. **D, E** Immunofluorescence staining showing colocalization of EBF3 (red) and SPiDER (green) in femoral bone sections. DAPI stained nuclei are blue and quantification of the percentage of EBF3 and SPiDER double positive cell area. Scale bar: 50 μm. $n = 4$ mice (Veh), 3 mice (PTX), and 6 mice (PTX + D/Q). *$P < 0.0116$; **$P < 0.0069$; $P = 0.9992$, ns = not significant. **F, G** Analysis of tumor burden in the femurs and tibias by bioluminescence imaging (BLI) after 13 days. $n = 9$ (Veh), 9 (PTX), and 10

(PTX + D/Q). 'n' indicates the number of paired intact femurs and tibias. *$P = 0.0133$; **$P = 0.0160$ (Veh $vs$ PTX + D/Q); $P = 0.9967$, ns = not significant. **H, I** Representative μCT images and quantitative analyses of trabecular bone volume to total volume (BV/TV). Scale bar: 100 μm. $n = 5$ mice (Veh), 6 mice (PTX), and 6 mice (PTX + D/Q). **$P = 0.0026$; *$P = 0.0242$. **J–L** Representative pictures of TRAP staining of osteo-clasts. Insets show a magnified view of osteoclasts (arrows) on the trabecular bone surface. Tumor areas are outlined with black circles for visual reference. Scale bar: 100 μm. Quantification of the number of osteoclasts to bone surface (N.Oc/BS) and osteoclast surface to bone surface (Oc.S/BS). $n = 8$/group. ****$P < 0.0001$; ****$P < 0.0001$ **K, L.** Data are represented as mean ± SEM and significance was determined by one-way ANOVA with Tukey test. Source data for this figure are provided as a Source Data file.

senescent cells are characterized by a minimal mitochondrial outer membrane permeabilization (miMOMP)[51] that makes them exceed-ingly sensitive to reductions in Bcl-2 family members or increases in the pro-apoptotic factors such as Bid, Bax, Bak and Bad. These observations led to the development of senolytics, including dasatinib plus quercetin (D + Q) that can effectively reduce senescent cell bur-den in mice[52]. Our present study showed that we can ameliorate chemotherapy-induced bone loss by depleting senCARs and senB-MAds using D + Q without diminishing the anti-tumor effects of che-motherapy, raising the possibility that senolytics could be an important treatment for patients. However, because tumor within the bone can also drive bone loss, we cannot rule out that the senolytics did not have a small effect on tumor-induced bone loss. Tumor induced bone loss also occurs in animals harboring only primary tumors, indicating that systemic signals can drive bone loss[53]. Given chemotherapy-induced bone loss results from senescent bone resi-dent cells, we did not examine the effects of senolytics in animals with primary tumors. Finally, caution needs to be taken as the timing of senolytic treatment may impact its effectiveness. Indeed, a recent study by Ali et al. showed that senolytics failed to rescue chemotherapy-induced bone loss[54]. Interestingly, in this study, the authors failed to detect senescent cells after four weeks of treatment, which is consistent with our previous findings[6]. These findings emphasize that early intervention is likely essential for the efficacy of senescence-targeted therapies in preserving bone mass following chemotherapy.

### Limitations of our study
Our data clearly demonstrate that chemotherapy induces p16[+] senes-cence specifically in bone resident CAR cells and BMADs. What remains unclear is why senescence is restricted to these cells. Regardless, our data show that senCARs and senBMAd express RANKL that drives bone loss. However, it is unclear if these two cell types contribute equally to bone loss or if one cell type is dominant. Regardless, our findings uncover the mechanism by which chemotherapy drives bone loss and provide two possible therapeutic approaches.

## Methods
### Animals and treatment
All animal experiments were performed in compliance with Washing-ton University in St. Louis's Animals Studies Committee. All animal procedures were approved by Washington University's Institutional Animal Care and Use Committees (IACUC). Mice were maintained on a 12-hour light/12-hour dark cycle, with ambient temperature between 20–22.2 °C and relative humidity of 30–70%. Mice were maintained on a regular chow diet (PicoLab Rodent Diet 20, Cat. No. 5053) formulated with 20% protein. C57BL/6 J (JAX, #000664) and albino C57BL/6 J mice (JAX, #000058) mice were purchased from the Jackson Laboratory. The INK-ATTAC and p16-Cre[ERT2]/tdTomato mouse models were gen-erated by Darren Baker's group[10] and Makoto Nakanishi's group[19], respectively. ADQ-Cre/DTR and ADQ-Cre/QR mice were generated by

crossing C57Bl/6 ADQ-Cre transgenic mice with DTR[f/f] and INK-ATTAC[f/f] (INK-QR) mice, respectively. AdipoCre[ERT2]/Rankl and AdipoCre[ERT2]/MK2 mice were generated to delete the gene of interest conditionally by crossing C57Bl/6 ADQ-Cre[ERT2] transgenic mice with Rankl[f/f] and MK2[f/f] mice, respectively. Mice harboring the AdipoCre[ERT2] transgene were fed tamoxifen to induce Cre recombinase activity. In this study, both female and male animals were included in the experiments unless otherwise specified. For tumor studies, a meta-static tumor model was established using PyMT-BO-1 (GFP/Luc) cells in albino mice. Briefly, 50 μl of a single-cell suspension containing 50,000 PyMT-BO-1 (GFP/Luc) cells was injected directly into the left cardiac ventricle (intracardiac injection) under anesthesia. On day 2 post-injection, metastatic lesion formation was evaluated and confirmed by in vivo bioluminescence imaging (BLI). Mice were then randomized into different treatment groups. Paclitaxel (PTX, 50 mg/kg) was administered intravenously (I.V.) on days 4 and 5, followed by a com-bination of dasatinib and quercetin (D + Q) administered every other day for a total of four doses. On day 13, both in vivo and ex vivo BLI were performed to assess metastatic tumor burden, and mice were subsequently sacrificed for tissue collection and further analyses. Experimental endpoints were determined by clinical signs associated with metastatic burden, including weight loss exceeding 20% of body weight. All mice were maintained in groups of three to five animals per cage with water and food *ad libitum*. Mice were euthanized using $CO_2$. The intact femurs and tibias were removed for further analysis. Dox-orubicin (LC Laboratories) was prepared in water and administered in a single dose at 5mg/kg via intraperitoneal (i.p.) injection, and paclitaxel (TOCRIS, a Biotechne brand) was administered in two doses at 50mg/kg via tail vein (IV). AP20187 (Batch No.: A113-20, Chemvada Life Sci-ences, San Diego, CA) and prepared with 100% ethanol: polyethylene glycol 400: 2% Tween-20 in molecular water at 4:10:86. AP20187 was administered for a week at 10mg/kg thrice through i.p. injections. Mice used in this study were genotyped by TransnetYX (an automated genotyping company, USA). Mouse details including strain, source/catalog number and injected compounds are provided in Supple-mentary Table 1.

### Bone marrow transplantation
Bone marrow transplantation was performed according to previously described protocols[5]. Recipient C57BL/6 CD45.1 mice received two doses of 400 cGy 4 hours apart, followed by transplant of bone mar-row by retro-orbital injection (i.v.). Irradiation was carried out using an X-ray irradiator (XRAD 320). Donor bone marrow was prepared from INK-ATTAC CD45.2 mice as follows: donor mice were sacrificed by $CO_2$ inhalation, both femurs, tibias and ilia were extracted in a sterile set-ting and flushed using pulsed centrifugation to collect marrow. Bone marrow was reconstituted in cold sterile serum-free 1X HBSS and injected retro-orbital at a concentration of 5 million cells per 200 μl per mouse. Mice were monitored over 2 weeks for signs of radiation sickness. 6-week post irradiation, mice were given paclitaxel and sacrificed 9 days later.

## Vertebral body transplant (vossicle) implantation

Vertebral bodies were harvested from 4-day-old INK-ATTAC pups. Soft tissues were stripped, and the vertebrae were sectioned into single vertebral bodies (Lumbar 4 and 5) with a scalpel blade. Female 11-week-old C57BL/6 J mice used as transplant recipients were anesthetized with isoflurane inhalation. Two 1-cm subcutaneous incisions and pouches were made at both sides of the flank region of each mouse. L4 and L5 were washed with PBS and implanted in the right and left pouches of each mouse, respectively. The surgical sites were closed using a surgical stapler.

## μCT scanning and assessment of the femur

Femurs were harvested from euthanized mice and after stripping the skin, the femurs were fixed with 10% neutral buffered formalin (NBF) overnight at 4 °C with gentle rotation. The bones were washed in phosphate buffered saline (PBS) and stored in 70% ethanol to be used for μCT. For μCT analysis, bones were suspended in 2% agarose and scanned using microcomputed tomography (μCT 50, Scanco Medical)) at 70kVp, 114μA, and 20μm resolution. For the trabecular compartment, contours were traced on the inside of the cortical shell using 2D images of the femoral metaphysis. The end of the growth plate region was used as a landmark to establish a consistent location for starting analysis, and the next 100 slices were analyzed. The following trabecular parameters are reported for all μCT experiments: bone volume over total volume (BV/TV)), bone mineral density (BMD), trabecular number (Tb.N), trabecular thickness (Tb.Th), and trabecular separation (Tb.Sp). For the cortical compartment, contours were traced on the outside of the cortical shell using 2D images of the femoral mid-diaphysis and 50 slices were analyzed. 3D reconstructions right below the growth plate of the femur were generated using Scanco μCT 50 Ray Tracer software.

## Bone histomorphology and tartrate-resistant acidic phosphatase (TRAP) staining

At sacrifice, mice femurs were isolated and fixed in 10% NBF for overnight. Bones were decalcified in 14% EDTA (Sigma, E5134), pH 7.2 for 14 days and embedded in paraffin. 5 μm sections of proximal metaphysis were prepared using a microtome (Leica, RM2235). Standard H&E and TRAP staining techniques were used for all bone sections. Images were collected using the Zeiss Axio Scan Z1 Brightfield slide scanner. To quantify the TRAP-positive cells and determine how much of their surface area covered the bone, we used the BioQuant software to quantify osteoclasts per bone surface (Oc.S/BS and N.Oc/BS). We carried out these measurements by defining a region of interest with the stamp tool window. We then chose a diagonal line type and set the distance to the growth plate and measured the N.Oc/BS and Oc.S/BS.

## Double bone labeling

Mice were administered calcein (10mg/kg, Sigma #C0875) and alizarin red (30mg/kg, Sigma #A3882) via i.p. injection 9 and 2 days prior to sacrifice, respectively. Femurs were fixed with 70% ethanol overnight at 4 °C with gentle rotation, then embedded in methylmethacrylate (MMA) for sectioning. Sections were left unstained and images were collected using the Zeiss Axio Scan Z1 fluorescence slide scanner, WUCCI. Histomorphometric analysis was performed using BioQuant software to quantify bone formation rate per bone surface (BFR/BS).

## Single-cell RNA-sequencing

12-week-old mice were treated with Veh- or Doxo (10 mg/kg) and sacrificed at 96 hours. Bones (femurs and tibias) were harvested from euthanized mice and placed in cold DMEM/F12 (Gibco). Muscle and connective tissues were removed. Bones were gently ground and then cut into small fragments. Fragments were transferred and digested with fresh 2 mg/ml collagenase (Sigma, C0130) in DMEM/F12 on a

rotating water bath at 37 °C for 30 min. Following the first digestion, the released cell suspension was filtered through a 70-μm nylon mesh into a collection tube and placed on ice. A second digestion of the remaining fragments was performed and the cell suspension was filtered into the same collection tube. Reactions were quenched with FACS buffer (PBS plus 0.5% BSA) with 2 mM EDTA. Following FACS buffer wash and RBC lysis, dissociated cells were labeled with anti-CD45 magnet microbeads (Miltenyi Biotech) and enriched into a $CD45^+$ fraction and $CD45^-$ fraction by MACS (magnetic-activated cell sorting, Miltenyi Biotech). Cells were stained with fluorochrome antibodies for 20 min on ice for cell sorting. We used CD45 (BioLegend, 103126), CD71 and Ter119 to deplete $CD45^+$ $CD71^+$, $Ter119^+$ populations using a BD FACS Aria III Cell Sorter. After sorting, cells were delivered for library construction using 10X Genomics Chromium 3′ GEM Single-Cell Library v3 kit (10X Genomics). Sequencing was performed according to a standard pipeline at The Genome Technology Access Center at Washington University, St. Louis. The Cell Ranger Software Suite from 10X Genomics was used for sample demultiplexing, barcode processing, and single-cell counting. The Cell Ranger count was used to align samples to the reference genome GRCm38 (mm10). For data analysis, the filtered feature barcode matrices were imported into R studio and analyzed using Seurat[55].

Each Seurat object was filtered to exclude genes expressed by less than 3 cells, cells expressing less than 200 genes or more than 10,000 genes, cells with more than 10% mitochondrial RNA content and cells with less than 100 counts or more than 100,000 counts. Log-based normalization, identification of variable features and scaling was performed using the corresponding Seurat functions (NormalizeData, FindVariableFeatures and ScaleData). Next, principal component analysis (PCA) and imputation using adaptively threshold low-rank approximation (ALRA)[56] was performed on the object. UMAP dimensional reduction was performed using the first 35 dimensions. We then used the FindNeighbors and FindClusters functions to cluster cells at a resolution of 0.1. Canonical markers were used to identify immune ($Ptprc^+$) and endothelial clusters ($Epcam^+$, $Cdh5^+$). Bone cells that were not immune or endothelial cells were re-clustered and canonical markers were used to identify the different bone cells. Due to the small size of the pericyte cluster, it was excluded from the final object named "Bone cells". The final object was then carefully inspected for markers of senescence. DGE analyses between clusters were performed using MAST analysis[57] and the obtained gene list was pre-ranked for gene set enrichment analysis (GSEA) based on p-values and $\log_2$fold change. All visualizations of gene expression were performed using either the RNA or the ALRA assay of the data set and are indicated accordingly in the figure legends.

For the analysis of the publicly available human bone scRNA-seq dataset (GEO: GSE230295) from acute lymphoblastic leukemia patients[18], we utilized N = 4 patient samples, with peripheral blood samples excluded. As formatted in the original dataset, samples collected at diagnosis (day 0) and post-treatment (day 15) were pooled for each patient prior to analysis. A merged object comprising all four patient samples was generated, and $CD45^+$ hematopoietic cells were excluded by sub-setting cells with detectable $Ptprc$ expression. The remaining $CD45^-$ stromal populations were normalized and reclustered using 10 principal components and a resolution of 0.2. All downstream analyses and plots were performed using this $CD45^-$ object.

## Reverse transcription qPCR

To quantify gene expression, the marrow cavity was flushed out by centrifugation at high speed for 10 s at 4 °C. Bones devoid of marrow were lysed and homogenized in TRIzol (Invitrogen). RINO Bullet-Blender Navy Bead Lysis tubes and Bullet Blender® Homogenizer were used for crushing the bone thrice at 4 °C with high speed for 5 min. Once the bones were completely crushed, the tubes were centrifuged

at 9000 x g for 10 min at 4 °C. The supernatant was transferred to a new tube for RNA extraction. Total RNA was extracted using the RiboPure™ RNA Purification Kit (AM1924, Invitrogen). cDNA was prepared from RNA (1 µg) using the High-Capacity cDNA Reverse Transcription kit (Applied Biosystems) followed by preamplification. RT-qPCR was performed using the Taqman PrimeTime ® Gene Expression Master Mix kit and gene-specific primers and probes. All reactions were performed in duplicate. mRNA was normalized to the mean of two housekeeping genes (either cyclophilin with actin or TBP with tubulin). All primer sequences used in this study are listed in Supplementary Table 1.

### Senescence associated β-galactosidase staining

SA-β-gal staining in femurs of mice was performed on frozen sections that were fixed in 10% NBF overnight at 4 °C with gentle rotation. Next day, following PBS wash, decalcified in 14% EDTA (pH 7.2) for 3 days at 4 °C with gentle rotation. Then, placed in a solution of 30% sucrose overnight. Next day embedded in OCT compound (Fisher Health Care, 4585) and sectioned 10 µm using cryotome (Leica, CM1950). Slides were washed in PBS for 2 min and then submerged in X-gal solution (1 mg/ml 5-bromo-4-chloro-3-indolyl β-D-galactopyranoside, 150 mM NaCl, 2 mM MgCl2, 5 mM K3Fe(CN)6, 5 mM K4Fe(CN)6, 40 mM NaPi pH 6.0, in H$_2$O). The X-gal solution was passed through a 0.22 µm filter prior to use in order to remove particulate. Slides were kept at 37 °C in the dark until the stain developed (~6 h). After staining, slides were washed in PBS, and then the nuclei were counterstained using Nuclear Fast Red (Sigma). Slides were scanned by Zeiss slide scanner and SA-β-gal positive area was analyzed by HALO software.

### Isolation of osteocyte-enriched cells

Briefly, mouse femurs and tibias were harvested and stripped of soft tissues followed by crushing. Subsequently, digested with fresh 2 mg/ml collagenase twice at 37 °C for 30 min. The remaining cell fraction represents a highly enriched population of osteocytes[8], which was used for gene expression analysis.

### Immunofluorescence staining (frozen sections)

Harvested bones were fixed in 10% NBF overnight and decalcified in 10% EDTA for 3 days at 4 °C followed by dehydration with 30% sucrose. Bones were embedded in OCT compound and femurs were cut longitudinal at 10 µm using a cryotome. Sections were washed, permeabilized for 10 min in 0.2% Triton-X in PBS and blocked for 1 h at room temperature using 0.5% BSA (Sigma, D9663) with Fc blocker (1:200) prior to incubation for overnight at 4 °C with rabbit anti–p16 antibody (1:100, Cat. No. PA1-46220, Invitrogen) and mouse anti-PPARγ (1:300, Cat. No. 66936-1, Proteintech). The specificity of the p16 antibody was evaluated prior to use in p16-knock out mice femur (Fig. S2I). Next day, sections were washed with TNT buffer (0.1 M Tris–HCL pH 7.4, 0.15 M sodium chloride, 0.1% Tween-20) followed by incubation with secondary goat anti–rabbit Alexa Fluor 568 antibody (1:400, Cat. No. A-11011, Invitrogen) and goat anti–mouse Alexa Fluor 488 antibody (1:500, Cat. No. A-11001, Invitrogen). All sections were stained and mounted using SlowFade Gold antifade reagent with DAPI (Invitrogen S36939) according to the manufacturer's instructions. Mouse on mouse blocking was performed when anti-mouse antibody was used according to manufacturer instruction (Vector Lab. PK-2200). All fluorescent images were captured and scanned using a Nikon Eclipse 90i microscope or Zeiss Axio scan Z1 fluorescence slide scanner. Osteocalcin$^+$ (OCN$^+$) osteoblasts were analyzed using HALO software, with specific attention given to accurately quantifying bone surface-associated OCN$^+$ cells. To ensure the precision of our analysis, OCN$^+$ signals located within the marrow cavity were carefully excluded. However, OCN$^+$ signals in close proximity to the bone surface were included. Importantly, the same criteria and analysis parameters were uniformly applied across all treated and untreated groups to ensure

consistency and comparability. Antibodies details including catalog number and identifiers are listed in Supplementary Table 1.

### Isolation of bone marrow adipocytes

Mature adipocytes (BMAds) were isolated directly from the bone marrow of mice according to previously described protocols[21,22]. Briefly, femurs were collected from mice, and the two ends of the bones were snipped. The bones were placed in a small microcentrifuge tube (0.5 ml) that was cut open at the bottom. The small tube with the bones was then placed into a bigger microcentrifuge tube (1.5 ml). Fresh bone marrow was spun out by quick centrifuge and resuspended in PBS. After centrifugation (900 x g, 5 min, RT), floating adipocytes were collected from the top layer and washed with PBS for 3 times.

### In Vitro differentiation of bone marrow cells

BMSCs were isolated from the femurs and tibias of 12-week-old wild-type mice treated with Veh or Doxo. After 9 days of treatment, the femurs and tibias were harvested and the proximal ends of the bones were cut followed by isolation of BM with the aid of a syringe. To check the differentiation potential of BMSC, we cultured isolated BMSCs directly in osteogenic or adipogenic induction media. For osteogenic induction, BMSCs were cultured in 12-well plates at a density of 1×10$^4$ cells/well. The medium was changed every 3 days. After culture for 14 days, the medium was removed, and cells were washed twice with PBS, then alkaline phosphatase (ALP) staining was performed where alkaline phosphatase substrate BCIP/NBT (5-bromo-4-chloro-3-indolyl phosphate/nitro blue tetrazolium) (SigmaFast, B5655) was added and incubated for 30 min at room temperature (RT). For alizarin red staining, after culture for 21 days, the medium was removed, and cells were washed twice with PBS, then fixed with 4% paraformaldehyde at RT for 30 min, after which they were rinsed twice with PBS. Finally, cells were stained with alizarin red s dye (Sigma, A5533) for 30 min and then rinsed thrice with distilled water for the removal of unbound stain, after which stained calcium nodules were captured by inverted microscopy. For adipogenic induction, BMSCs were seeded in 12-well plates at a density of 1 × 10$^4$ cells/well and at day 14 images were captured. For lipid droplet staining, BMSCs were seeded on coverslip (22 x 50mm) at a density of 1×10$^4$ cells and at day 14, cells were washed twice with PBS and fixed with 4% paraformaldehyde at RT for 30 min. They were then washed twice with PBS and stained with 1 µg/ml of lipophilic fluorochrome BODIPY 493/503 (Invitrogen, D3922) for 15 min at RT. They were then washed thrice with PBS, mounted using SlowFade Gold antifade reagent with DAPI and stained lipid droplets were scanned using the Zeiss Axio Scan Z1 fluorescence slide scanner, WUCCI.

### Oil Red O Staining

Frozen sections were allowed to air dry at room temperature for 1 h and washed with water for 5 min followed by 60% isopropanol wash for 15 sec. Oil red O (Sigma, 00625) prepared according to the supplier was added and incubated for 15 min with gentle shaking in the dark. After incubation, rinsed with 60% isopropanol for 15 sec followed by hematoxylin staining and then, mounted using permanent mounting medium. For SA-β-gal/Oil red O-double staining, frozen sections were first immersed in SA-β-gal solution and incubated for 5–6 h at 37 °C. After that, slides were washed and stained with oil red O staining. Oil red O-stained positive cells or SA-β-gal/oil red O-double positive cells were visualized and captured using an inverted microscope (Nikon Eclipse 50i) equipped with a Nikon DS-Fi3 camera with 20x magnification. SA-β-gal/oil red O-double positive area were analyzed by HALO software.

### Oral feeding of p38i/MK2i compounded chow

The p38MAPK small-molecule inhibitors CDD111 (p38i) and CDD2231 (MK2i) (Aclaris Therapeutics, Inc.) were compounded at 1000 PPM. Both were compounded into Research Diets Inc., catalog number

5001. Female C57BL/6 mice were fed ad libitum. Mice were randomized onto inhibitor-containing or regular chow on the same day as their first doxorubicin dose till 9 days.

## IC injection and BLI

12-week-old female albino C57BL/6 J mice were anesthetized with 5 µl/g body weight of Ketamine/xylazine cocktail (17.7 g/ml of ketamine and 2.65 mg/ml of xylazine). When animals were completely anesthetized, 50 µl of PyMT-BO-1 (GFP/Luc) cells (50,000 cells) were injected directly into the left cardiac ventricle (IC injection). The PyMT-BO-1 cell line was gifted by Kathy Weilbaecher. Prior to use, we verified the PyMT-BO-1 cells using STR profiling (ATCC) and mycoplasma testing. Bioluminescence imaging (BLI) was carried out as previously described[58]. Ex-vivo imaging was performed on an IVIS100 or IVIS Lumina (PerkinElmer; Living Image 3.2, 1–60 sec exposures, binning 4, 8, or 16, FOV 15 cm, f/stop 1, open filter). Mice were injected with D-luciferin (150 mg/kg in PBS; Gold Biotechnology) via i.p. After being sacrificed, both hind limbs were harvested and imaged for 10 seconds. For analysis, total photon flux (photons/sec) was measured from a fixed region of interest (ROI) over the whole bones using Living Image 2.6 software.

## Statistics

All statistical analyses were carried out using Prism (latest version 10.4.0). Data are expressed as mean ± SEM. Between two group comparisons, unpaired two-tailed Student t-tests were used. For multiple comparisons, one-way ANOVA with Tukey test was used. A p-value of less than 0.05 was considered statistically significant. Quantitative assessments of µCT-based analyses were performed by an individual who was blinded to the sample identity.

## Reporting summary

Further information on research design is available in the Nature Portfolio Reporting Summary linked to this article.

## Data availability

Source data are provided with this paper. The scRNA-seq datasets analyzed in this study are available through the NCBI Gene Expression Omnibus (GEO) under the accession codes GSE289491 (samples derived from mice) and GSE230295 (samples derived from acute lymphoblastic leukemia patients)[18]. Raw data including micrographs can be found on Zenodo[59]. Source data are provided with this paper.

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

## Acknowledgements

We thank Roberta Faccio, Erica Scheller, Deborah J. Veis, Dan Link, Steven L. Teitelbaum, and Sundeep Khosla for constructive input. We are grateful to Steven L. Teitelbaum for providing the ADQ-Cre/DTR mice and Dan Link for the Cxcl12-GFP mice and advice. We thank Michael Brodt for his assistance with μCT imaging in the Musculoskeletal core, Washington University School of Medicine. We are also grateful for the core services provided by the Musculoskeletal Research Center (NIH P30-AR074992), the Washington University Center for Cellular Imaging (WUCCI, supported by the Washington University School of Medicine), the Siteman Cancer Center Flow Cytometry Core, and the Genome Technology Access Center (GTAC). We also thank Makoto Nakanishi group for providing the p16-Cre^ERT2 mice. This work was supported by NIH grants R01CA282810, R01AG059244, and R56AG088264 (S.A. Stewart), the U.S. Army Medical Research Acquisition Activity, 820 Chandler Street, Fort Detrick, MD 217025014, is the awarding and administrating acquisition office, and this was supported in part by the Office of the Assistant Secretary of Defense for Health Affairs, through the Breast Cancer Research Program, under award No. MBC181712. Opinions, interpretations, conclusions, and recommendations are those of the authors and are not necessarily endorsed by the Department of Defense. This work was also supported by the NCI Cancer Center Support Grant P30CA091842, Fashion Footwear Association of New York, and the Alvin J. Siteman Cancer Center, Siteman Investment Program (supported by The Foundation for Barnes-Jewish Hospital, Cancer Frontier Fund) to S.A. Stewart.

## Author contributions

Conceptualization, data curation, formal analysis, investigation, methodology, visualization, writing–original draft, writing–review and editing: G.K.R. Investigation: T.M., R.R.-O., X.Y., Z.Y., and D.V.F. Investigation and formal analysis: A.M. and T.H. Investigation and resources: Q.R. Resources: D.G.D. Conceptualization, resources, data curation, supervision, funding acquisition, writing– original draft, project administration, writing–review and editing: S.A.S. All authors approved and provided comments on the submitted manuscript.

## Competing interests

The authors declare no competing interests.
