## [Transparent Peer Review file · Nature Communications]

Chemotherapy-induced adipo-lineage cell senescence drives bone loss

Corresponding Author: Dr Sheila Stewart

Version 0:

Reviewer comments:

Reviewer #1

(Remarks to the Author)

Manuscript summary

This study provided mechanistic insights into how chemotherapy-induced senescence leads to bone loss. By using scRNA-seq and performing validation experiments on multiple genetically modified mice in non-tumor setting, the authors provided compelling results that chemotherapy-induced senescence is restricted to bone marrow adipo-lineage cells, and these cells contributed to bone loss through increased expression of Rank ligand, which led to increases in osteoclast number and activity. The authors also evaluated the effect of senolytic drugs in protecting against bone loss in tumor-setting (breast cancer), though further investigation is needed in the future. Overall, this work presented a novel finding that targeting senCAR/senBMADs may be a promising approach to prevent bone loss caused by chemotherapy. There are many strengths to this report, including complete and rigorous experiments and data analysis in vitro and in vivo. The findings have high clinical relevance. These data provide mechanistic insight. However, there are some limitations, detailed below, which should be addressed.

Questions

1. Can chemotherapy cause direct senescence in osteoclasts (OCs)?

Fig.1F shows increased expression of p16 in crushed bone devoid of bone marrow cells, which indicates that senescence occurred in cells on or in the bone. Fig 1J shows decreased OCs in doxorubicin (Doxo) group treated with AP20187 (AP, the drug that eliminates p16+ senescent cells). Considering AP eliminates p16+ cells systemically, can the decrease in OCs be due to the elimination of AP on senescent OCs?

2. The statement on “systemic senescent factors do not contribute to bone loss” (line 183-184) is overstated.

In lines 180-184, the author stated that “AP did not rescue bone loss and reduce TRAP+ OCs in WT, indicating that systemic senescence factors do not contribute to bone loss”. By using vossicle model (Figure 2), implanting vertebrate from INK-ATTAC mice in C57BL/6 WT mice, the observation that AP protected bone in INK-ATTAC vertebrate but failed to prevent bone loss in WT femur can only indicate that bone resident cells contribute to Doxo-induced bone loss. In order to assess the effect of systemic factors, reversed implantation (implanting vertebrate from WT mice in INK-ATTAC mice) will need to be performed. If similar bone loss was observed in WT vertebrates implanted in INK-ATTAC mice with Doxo and AP treatment compared to the group with Doxo only, it may be indicated that systemic senescence factors do not contribute to chemo-induced bone loss.

3. Clarification is needed for what type of sample is used for scRNA-seq.

Line 212 states that the author performed scRNA-seq on bone marrow. However, in Fig. 3A, a step in the schematic shows bone crushing, which causes confusion about whether whole bone (crushed bone + bone marrow cells) or only bone marrow cells should be used for scRNAseq. Based on the context, it is likely that the author used bone marrow cells. However, considering that senescence can occur in bone resident cells (see question 1), sequencing only on bone marrow cells rules out the possibility of senescent bone resident cells contributing to chemotherapy-induced bone loss.

4. Related to Figure 4

a. In lines 251-252, it states that “no significant changes in p16+/ppary- cells across the groups”. There are no images in Fig. 4 that support this statement.

b. In line 271, it states that “there were no changes in Ebf3-SPIDER+ cells in Veh and Doxo groups”. What about Ebf3-

;tdTom+ cells?

c. Fig. 4M needs a clear label. Based on the results description, these μ CT images were taken from DTRADQ mouse treated with PBS or DT. PBS should be labeled above the first paired μ CT images, and DT instead of iDTRADQ should be labeled above the second paired μ CT images.

5. Related to Figure 5

In line 339 and Fig. 5C-F, please clarify which bone samples were used for RT-qPCR, bone marrow, crushed bone (depleted bone marrow), or whole bone?

6. Related to Figure 6

a. Fig. 6M, IF staining of RANKL. Can the author include a description of the location of the stained RANKL in text? Is it primarily present in the bone marrow space or on the bone?

b. Line 412-413 states that the expression of other SNAP factors decreased after p38i/MK2i in crushed bone depleted bone marrow. Since senCAR/BMADs reside in bone marrow, expression of these SNAPs factors needs to be detected in bone marrow to further confirm the inhibitory effect of p38i/MK2i on SNAPs.

7. Related to Figure 7

a. Representative images are needed for Fig. 7B

b. For Fig. 7E-F. Bioluminescent images and intensity quantification are not sufficient to assess tumor burden. Please show histological images of tumor-bearing bones and quantify tumor burden by histomorphometry. Histomorphometric analysis is the gold standard for measuring tumor burden in bone.

c. Fig. 7G. How to distinguish bone loss caused by PyMT-BO-1 cells from paclitaxel (PTX)?

d. In Fig. 7J, tumors are not observed on the histological section, which is inconsistent with the bioluminescent images (showing tumor grew in bones). High-resolution images need to be provided in order to show tumor.

e. Given that (1) PTX and Doxo may have different anti-tumor effects on breast cancer, and (2) senolytic drugs (D+Q) may interact differently with chemotherapies, please provide justification for why PTX, not Doxo, was chosen to use in the breast cancer bone metastatic model.

Overall, the data presented in figure 7 is not of the same quality as previous figures. The rationale for using the bone metastatic model to support the authors' findings is not clear. By introducing tumors in the model, it complicates the results and makes it hard to evaluate the changes in bone due to tumors or chemotherapy. The author may consider excluding the figure 7 from the manuscript since the model used here is not appropriate, and the results do not align with or support the main conclusion. Future work could investigate the author's findings by using adjuvant breast cancer models.

8. Others

a. Detailed methods are needed on how the author defines and measures the OCN+ area on bone sections.

b. In line 271, typo. Fig. 4SL should be Fig. S4L

c. The phrase "bone resident cells" is used multiple times throughout the manuscript. The authors seem to use it to refer both to bone marrow cells and cells in/on bone, which causes confusion. Please clarify it in text or figure legend, as appropriate.

Reviewer #2

(Remarks to the Author)

Evaluation report for the manuscript entitled "Chemotherapy-induced adipo-lineage cell senescence drives bone loss" by GK Raut and collaborators submitted to Nature Communications.

GK Raut et al. previously reported that elimination of senescent cells and inhibition of SASP production using genetic and pharmacological approaches inhibiting the p38MAPK-MK2 pathway prevented chemotherapy-induced bone loss. In this report, by using an extensive number of mouse models, genetic and pharmacological approaches, the authors propose that chemotherapy-induced senescence does not apply to osteocytes, as previously reported by Sundeep Khosla's group (PMID: 27341653). Instead, the authors found that chemotherapy-induced senescence is limited to bone resident CAR cells (senCARs) and, also, bone marrow adipocytes (senBMADs), which crucially contribute to bone loss through RANKL ligand-driven osteoclast activation. In addition, chemotherapy-induced senescence reduces osteoblast differentiation and the bone mineralization capacity resulting in a decoupling process of bone formation and resorption. Interestingly, the authors provide evidence that genetic and/or pharmacological (dasatinib + quercetin) elimination of senCARs and SenBMADs ameliorated chemotherapy-induced bone loss without affecting the anti-tumour activities of chemotherapy. Moreover, senomorphic targeting of RANKL (described as a SASP factor) by using p38i and MK2i inhibitors prevented chemotherapy-induced bone loss. Altogether, GK Raut and collaborators propose a therapeutic strategy to preserve bone integrity and improve quality of life during chemotherapeutic cancer treatments.

This is a well-presented and voluminous article in a hot topic of potential clinical relevance. The experiments are solid and elegantly designed, and the results align with the interest for a broad audience. Overall, the conclusions seem supported by the experimental data, however the manuscript must address the conceptual advance of this research, including conflicting data with other groups, and provide further experiments to reinforce the conclusions before it can be recommended for publication in Nature Communications.

Main points

1. The authors must address the conceptual advance of this work for a journal like Nat Comm, as they previously reported that chemotherapy-induced senescence drives bone loss using the same models. i.e. doxorubicin- and PTX-mediated, INK-ATTAC mice, and showing that clearance of senescent cells rescues osteoblast function and reduces osteoclasts numbers

(Yao et al. Cancer Research 2020, PMID: 31932453). Also, in the same previous publication, the authors showed that the process of bone loss is driven by the p38MAPK-MK2 pathway, and senomorphic treatment with p38i/MK2i inhibitors limits SASP production and rescues chemotherapy-induced bone loss. Therefore, the novelty of the current study seems compromised and mainly stems from the identification of senCARs and SenBMAds as key players of the observed phenotype. In addition, it is well known that osteocyte RANKL is required for cortical bone loss with age and that it can be induced by senescence (Kim et al. JCI insight 2020, PMID: 32870816).

2. It seems concerning a recent publication from another laboratory (Ali et al. Scientific Reports 2025, PMID: 40389522) reporting that the adverse effects of chemotherapy on bone mass are not prevented by senolytics. In particular, because these authors use similar chemotherapies (i.e. doxorubicin, docetaxel and cyclophosphamide) and senolytics (i.e. dasatinib + quercetin and piperlongumine). Also, the readouts are fundamentally the same, like trabecular bone mass, thickness, etc. This point needs to be convincingly addressed. How do the authors explain this apparent discrepancy?

3. Following with the previous point, a possibility raised by Ali and collaborators is that the decrease in osteoblast numbers is due to chemotherapy DDR-induced osteoblast apoptosis. Therefore, mechanistic analyses in this manuscript must include a comprehensive characterisation of apoptosis (e.g. TUNEL or activated-caspase 3) damage makers (e.g. gH2AX) and how they relate to the crucial cellular types that maintain the intricate balance of bone damage/loss and formation/resorption.

4. The authors use the well-established INK-ATTAC mouse model to provide evidence that chemotherapy-induced senescence associates with disrupted bone homeostasis, and also p16CreERT2-tdTom mice to identify CARs and BMAds with p16 "senescent" features. To ensure specificity of the p16 antibody the authors tested bones from p16KO treated with Doxorubicin. If the authors are correct in their hypothesis, I find crucial to validate the main conclusions of this manuscript in chemotherapy-treated p16KO mice. What would be the phenotype and mechanism of this model in bone loss and senescence induction?

5. Also, the authors do not validate the potential role of p21 as senescence biomarker over the manuscript. In fact, the p53/p21 axis is an essential component of the chemotherapy-induced DDR response. Are senCARs and senBMAds positive for both p16 and p21? Or there are different senescent ecosystems? In this regard, distinct secretomes in p16 and p21 positive senescent cells have been proposed across tissues, including the bone (Saul et al. bioRxiv, <https://www.biorxiv.org/content/10.1101/2023.12.05.569858v1.full.pdf>).

6. Any validation with human datasets / samples post-chemotherapy regarding senCARs / senBMAds would reinforce the conclusions and translatability of this study, although I'm aware of the difficulties in getting such samples.

Other points

- The Discussion is currently underdeveloped and should address all concerns regarding novelty, contradictory data with other groups, and the limitations of this study.

- RNAseq analyses rely on SenMayo datasets, but it would be important to correlate the data with other datasets and senescent signatures.

Reviewer #3

(Remarks to the Author)

In the outlined manuscript, Ganesh Kumar Raut and colleagues describe in a compendium of in vivo experiments using several genetic mouse lines, that chemotherapy-induced bone loss is linked to the induction of senescence specifically in adipo-lineage cells of the bone. Adipo-lineage cells include CXCL12-abundant reticular (CAR) cells and mature adipocytes, both marked the expression of adiponectin in mouse bones. In previous findings from the group (Yao et al., doi: 10.1158/0008-5472.CAN-19-2348), they showed that chemotherapy induced bone loss can be prevented by the ablation of senescent cells using a Cdkn2a driven suicide gene (INK-ATTAC) or the use of senormorphics. In the previous publication they used a 28-day exposure to Doxo or PTX, showing that trabecular bone loss is due to loss of trabecular numbers and thickness, that aligns with an increase in osteoclast numbers and decreased bone formation, all of which was reversible through ablation of senescent cells.

In the current study the group expands these observations by the identifying the putative cell type, i.e., cells of the adipo-lineage, being susceptible to the induction of senescence which in turn drives the expression of RANKL and consequently osteoclast mediated bone loss. However, the authors do not provide regulatory mechanisms, i.e., which signals, and pathways confer the high susceptibility of CAR cells and adipocytes to chemotherapy treatment, and whether those signals directly or the SASP gene signatures drives expression of TNFSF11 specifically in adipo-lineage cells. In addition, the authors do not provide information whether baseline adipocyte content, i.e., marrow adiposity due to ageing, obesity, or caloric restriction, has an impact on the degree of chemotherapy induced bone loss. Taken together, these new insights are limited in the scope of Nature Communications and its broad audience.

While depletion of adiponectin expressing cells or TNFSF11(RANKL) ablation specific to adiponectin expressing cells convincingly shows to prevent chemotherapy induced bone loss, the evidence of the senescence signature being restricted to CAR cells and adipocytes is less supported. The latter is mainly due to the usage of a variety of genetic mouse models or staining procedures which are not directly comparable (often using several methods to stratify the adipo-lineage but using a single approach to deny the role of other bone resident cells). This is surprising as the authors performed experiments, such as the adipocyte fractionation and scRNA-sequencing in which different compartments could have been compared using the same method to underline the susceptibility of adipocyte-lineage cells to senescence.

In addition to that, the phenotype of trabecular bone loss characterized in supplementary figure 1 is barely recapitulated throughout the intervention studies and often bone loss monitored through loss of BV/TV does not correspond to changes in trabecular numbers or thickness, questioning the cause of bone loss.

Major points:

Trabecular bone loss has been initially characterized by loss of trabecular numbers, however this parameter is not changed throughout the intervention studies in the INK-ATTAC mice, adipocyte depleted mice, mice lacking RANKL in adipocytes, or mice treated with senolytics or senomorphics. In addition, the effects of trabecular thickness are limited and therefore are unlikely to explain the observed bone loss. Did the authors check mineral content and density to explain the loss of trabecular bone without effects on trabecular numbers and thickness.

The authors use a lot of space for the clustering the cell-type identification in the scRNA-seq data, but do not follow up on this throughout the manuscript. Therefore clustering could be part of the supplement. In contrast, the gene set enrichment analysis should be performed by comparing Doxo condition to vehicle for each individual cluster to highlight that CAR cells are more susceptible to the induction of senescence than the other cells. For now, the authors compare CAR cells with the remaining cells, but not the induction levels in each cluster individually. A heatmap of the Sen-Mayo gene signatures could be additionally used to quantify changes in senescence associated gene transcription.

Along the line, the fractionation of mature adipocytes for gene expression (Fig. 4D) should be complemented by analysis of the SVF fraction, i.e., the cells in the pellet, to compare induction levels of SASP factors across fractions.

While the authors emphasize to outline adipo-lineage cells to be highly susceptible to chemotherapy induced senescence and causally involved in bone loss, they do not show how adipocyte ablation and TNFSF11 conditional knockout are related to the senescence burden within the bone. Thus, estimation of senescent cells through P16 or SPIDER staining should be performed in the mice iDTR-ADQ mice to prove the loss of senescent cells as well as in TNFSF11-AdipoCRE mice to prove the persistence of senescent adipo-lineage cells.

Data availability statement is lacking describing the deposition of raw as well as processed data. Making their data set accessible, especially a well-annotated Seurat object would have a big impact on the implementation of the data in future studies.

Minor points:

The panels characterizing the histological data, fluorescence as well as H&E or TRAP staining are often way too small.

Quantification of osteoblasts (1M,S5D, 6I) should be done through measuring osteocalcin positive cells on the bone surface and not the total area of staining throughout the marrow space.

The in vitro differentiation experiments are misleading, as the isolated cells of Doxo treated mice do not survive or do not proliferate to the extent of cells from vehicle treated mice. These data should rather be replaced with proliferation curves and CFU-assays but also be part of the supplementary figures.

Is it correctly described that the gene expression analysis in Supplementary Figure 7J to L has been performed on crushed bones depleted for bone marrow? Wouldn't this imply that the material is devoid of both CAR cells and adipocytes, but still the authors find induction of SASP members? How does this relate to the fact that senescence is restricted to adipo-lineage cells?

The authors should implement a time line for the diphtheria toxin and tamoxifen treatments, i.e., how many days after adipocyte or RANKL ablation do they initiate Doxo or PTX treatment. This would help to understand the timing to establish the basal differences in the iDTR-ADQ and TNFSF11-AdipoCRE mice to their respective controls.

Color scale in Fig3G is not optimal, since the CAR cells 3 could have a huge white circle for Cdkn2a which is not visible. Since the color bar indicates increasing levels, making the color matrix going through white is not optimal.

The authors should state the house keeping gene in the figure panels as the method section indicates that different ones have been used.

The TNFSF11 induction in the scRNA-seq data should be rephrased, as the majority of CAR cells show down-regulation of TNFSF11 while a new population of cells with very high expression levels appear. In addition, a summed score of the Sen-Mayo gene signature could be used to correlate TNFSF11 expression levels. The latter can demonstrate that CAR cells with this very high TNFSF11 expression levels are also high in senescence-related gene expression.

The authors could add a statement on the limitation of the study, i.e., not being able to de-tangle the effect of CAR cells and mature adipocytes. In addition, the study aims to prevent bone loss but did not test whether senolytics or interference with adipocyte derived RANKL expression after established bone loss can restore bone over time.

The observation that TNFSF11-AdipoCRE mice have higher bone mass and loss of osteoclast on trabecular bone surfaces raises the question whether adipo-lineage derived RANKL is important for normal bone homeostasis and not only under pathological conditions, as the authors refer to (line 342). How is site-specificity obtained in a model in which cells distant to the bone surface control osteoclast maturation?

If not addressing in the future, the authors should discuss whether baseline adiposity in the marrow cavity affects chemotherapy induced bone loss, and if site-specific bone loss corresponds to local adipo-lineage content.

Finally, there is a trend of inconsistency of how to convey the main message, exemplified by the senolytics and senomorphics treatment. In the first senescence is estimated through the SPIDER assay and co-staining for EBF3 to focus on CAR cells, while with senomorphics the authors show P16 staining. The first intervention is supported by osteoblast numbers while the latter is through bone formation assays. The effect on RANKL levels are only documented for the senomorphics. This style appeals as existing data have been compiled. For that I would suggest to focus on common measures in the main figures, but this is clearly a question of taste.

Reviewer #4

(Remarks to the Author)

Reviewer #5

(Remarks to the Author)

Version 1:

Reviewer comments:

Reviewer #1

(Remarks to the Author)

The authors have addressed most of my concerns thoroughly. The quality of results related to the bone metastatic model (Figure 8) has been improved. I thank the author for adding the histomorphology assessments to support their findings. One additional point is that I suggest the author include their pilot experiment on testing the effects of senolytics on preventing chemotherapy-induced bone loss using the primary tumor model in supplementary data, as mentioned in their response to my question 7C. And, justify the reason for choosing the bone metastatic model over the primary tumor model. In fact, I think a non-metastatic breast cancer model is more suitable for testing their hypothesis than a bone metastatic breast cancer model, since it is tricky to distinguish bone destruction caused by chemotherapy or metastatic cancer. Although the author mentions a paper (Pin et al., Cancer Lett 2021) states that primary tumors can also cause bone loss through systemic endocrine effects, the results from the paper were not obtained using breast cancer model.

Reviewer #2

(Remarks to the Author)

The authors have now sufficiently addressed my concerns. I have no further comments and congratulate the authors for the work done..

Reviewer #3

(Remarks to the Author)

Ganes Kumar Raut et al. did a great job in answering the technical as well as conceptual concerns raised during the revision of the manuscript. Importantly, in the revised manuscript the authors made use of their existing data to show that Car cells are especially susceptible to the induction of a senescence gene signature along with induction of Tnfsf11 expression. In addition, they also follow up on the presence of senescent cells when interfering with adipocyte numbers or Tnfsf11 gene expression.

Major:

The main finding of the study only provides limited advantage to the field. As outlined in the rebuttal letter, the mechanistic insight is given by the identification of CAR and adipo-lineage cells as main cell type being responsible for chemotherapy-induced bone loss through a senescence-driven induction of RANKL production. While the authors argue with an important statement of: "These results reveal that even within the same tissue (i.e., the bone), the nature of the insult determines the cell type that senesces and the downstream consequences, highlighting, not diminishing the broader biological and clinical significance of our findings. Together, our findings and Khosla's work set a standard for the field by saying that simply showing that phenotypes get better upon the elimination of senescent cells without identifying the cell type or mechanism responsible is no longer sufficient." One needs to be aware that the induction of senescence specifically in bone marrow adipocytes is linked to glucocorticoid-induced bone loss (<https://doi.org/10.1016/j.cmet.2023.03.005>). The paper by Liu et al. also showed that transfer and elimination of senescent adipocytes can induce and rescue bone loss, respectively, already demonstrating the compartmentalization of senescence within bone tissue. Despite its potential clinical relevance, and the

clear and thorough execution of experiments, I judge the conceptual novelty to be limited. As such, conceptual novelty would be the identification of molecular cues driving the elevated susceptibility of CAR cells and bone marrow adipocytes to chemotherapy induced senescence to understand compartmentalization of senescence within the bone tissue. Or how senescence itself triggers RANKL expression and whether this is distinct from Tnsf11 induction during normal bone remodelling or auto-immune induced bone loss. Importantly, I recognize the strength of the data and rigor of the presented study, and I feel comfortable with the editor's decision to consider this manuscript for publication in nature communications.

Minor:

Conceptually, the in vitro differentiation data do neither support the main story line nor justify the conclusion taken in the manuscript of an impaired osteoblast differentiation capacity if cell numbers are the primary problem. As outlined in the revision, the authors claim that their data show impairment in proliferation, which is neither the conclusion in the main text nor supported by the assays as increased cell death could have a similar outcome. Those data might be removed from the manuscript.

The figures have random styles of colours and shapes, which does not compromise the scientific value or message, but might lead to confusion if use of colours is intended. Not limited to, but for example Fig 2C versus Fig 2F/2G and S3A-E,M or O for the INK-ATTAC mice or Fig 5B versus S7M for the ADQ-CRE/DTR mice.

Line 314, it should refer to Figure 5C and 5D instead of 4C and 4D.

Reviewer #4

(Remarks to the Author)

Reviewer #5

(Remarks to the Author)

We sincerely appreciate the reviewers' thoughtful evaluation of our manuscript. Their constructive feedback has been invaluable in strengthening the quality and clarity of our work. In response, we have carefully addressed each comment in a point-by-point manner that clarifies our findings and further supports our conclusions. However, as we read through the critiques, we feel the main concerns with the work surrounds questions of novelty and the clinical relevance. Given this, we have addressed these two main critiques immediately below.

A question of novelty: Our previous work established that senescence was involved in chemotherapy-induced bone loss and that a p38MAPK inhibitor could prevent this effect. However, at that time, we could not determine the specific cell type undergoing senescence, its location, or the mechanism by which it drove bone loss. The possibility remained that the p38MAPK inhibitor's effect was mediated through limiting osteoclastogenesis rather than impacting senescence (SASP) per se. Our current manuscript addresses these fundamental gaps by using complementary genetic and pharmacologic approaches.

The senescence field has been plagued with papers showing that by treating mice with senolytics or activating the suicide gene in the INKATTAC mouse, things got "better". Indeed, until our recent studies showing that senescent fibroblasts impact different immune cells based on tumor/tissue context (Ye et al., Can Disc 2024; Belle et al., Can Disc 2024), almost no paper showed a specific senescent cell target was cleared by senolytics or in the INKATTAC mouse. An exception was actually in the bone where Khosla's group showed that osteocytes senesce with age to drive bone loss. This is distinct from what we find with chemotherapy where it is CAR cells and mature adipocytes that senesce (CAR/BMAd). These results reveal that even within the same tissue (i.e., the bone), the nature of the insult determines the cell type that senesces and the downstream consequences, highlighting, not diminishing the broader biological and clinical significance of our findings. Together, our findings and Khosla's work set a standard for the field by saying that simply showing that phenotypes get better upon the elimination of senescent cells without identifying the cell type or mechanism responsible is no longer sufficient. Here, through the development of numerous advanced mouse models and extensive experimental approaches, we have shown that:

- in response to chemotherapy, senescence is induced within 9 days and contributes to bone loss.
- chemotherapy-induced senescence is restricted to CAR/BMAd.
- senescent bone resident CAR/BMAd are sufficient to drive chemotherapy-induced bone loss.
- activation of the p38MAPK pathway specifically in CAR/BMAd drives SASP and subsequent bone loss.
- a SASP factor (RankL) produced by senescent CAR/BMAd mediates bone loss.

A question of clinical relevance:

Chemotherapy-induced bone loss is an underappreciated but significant problem, as cancer survivors across tumor types face elevated fracture risks and increased long-term mortality. Our work uncovers the mechanism responsible for this adverse effect of chemotherapy, providing a direct translational pathway. We show:

- for the first time that senolytics, which are already in clinical trials for nonlethal conditions and have low toxicity profiles, can prevent chemotherapy-induced bone loss.
- importantly that administration of senolytics does not impair the anti-tumor efficacy of chemotherapy, addressing a crucial concern for integration into oncology practice.
- senescent CAR cells are present in human bone marrow following chemotherapy treatment.

In summary, our study provides the first detailed cellular and mechanistic insight into chemotherapy-induced bone loss and offers a clinically translatable intervention using senolytics—without compromising cancer treatment efficacy. We thank the reviewers for the opportunity to clarify and strengthen the manuscript in these respects.

Below, we continue with our point-by-point response to the reviewers' critiques.

Reviewer #1 indicated that our work provided mechanistic insights into how chemotherapy-induced senescence leads to bone loss. Further, Reviewer 1 highlighted that we provided compelling results that demonstrated that bone marrow adipo-lineage cells drive chemotherapy-induced bone loss. Finally, the reviewer indicated that additional strengths included complete and rigorous experiments and data analyses, mechanistic insights, and high clinical relevance. Below we discuss points raised by the reviewer.

Question 1. Can chemotherapy cause direct senescence in osteoclasts (OCs)?

Response: This is an interesting question but our work indicates that OC do not senesce. Examination of bone sections from p16-Cre^{ERT2}/tdTom mice revealed few tdTom⁺ cells on the trabecular bone surface where osteoclasts are located and no difference between vehicle or chemotherapy treated mice was observed (**Fig. S6J & S6K**). In addition, using our ADQ-Cre/Rankl^{f/f} mice, we see clear evidence of p16⁺ senescent cells (**Fig. S10P & S10Q**) yet OCs are reduced. Because OC do not express adiponectin (so cannot be killed), these data support our conclusion that the reduction in OC is due to loss of the SASP factor, Rankl (**Fig. S10M-S10O**). To further add to the rigor of our findings, we also created a cell type specific INK model (**Fig. S8A**) that we call the INK-QR mouse. The INK-QR mouse contains a floxed INK allele that allows us to activate the suicide INK gene in a cell type specific manner. We mated the INK-QR mouse to the ADQ-Cre mouse and found that we prevent bone loss (**Fig. 5C & 5D**), demonstrating that senescent ADQ⁺ cells drive bone loss, not OC.

Question 2. The statement on “systemic senescent factors do not contribute to bone loss” (line 183-184) is overstated.

Response: Our data show that bone loss is prevented exclusively in bones expressing the INK allele despite those bones being implanted into wild type mice whose femurs lose bone upon systemic chemotherapy treatment. If systemic factors not originating in the bone drove bone loss, we should not have prevented bone loss in the INK vertebrate. Thus systemic factors are not sufficient to drive bone loss. However, the reviewer does have a point, if the cells in the bone senesce in response to a systemic factor and those senescent bone cells drive bone loss, as we show, we would not have detected that in our system. However, given all of our data including the ADQ-Cre/iDTR (**Fig. 5A & 5B**) and now INK-QR mouse (**Fig. 5C & 5D**), our data

demonstrate that bone loss is driven senescent bone resident cells. However, to avoid confusion, we have removed reference to systemic factors in the updated manuscript.

Question 3: Clarification is needed for what type of sample is used for scRNA-seq.

Response: We apologize for not being more precise in the description of the samples. As shown in **Fig. 3A**, for the scRNA-Seq, we crushed and digested whole femurs (containing marrow) and then used MACs sorting to deplete CD45+, Ter119+, CD71+, and CD31+ cells and carried out scRNA-Seq on the remaining cells. This approach removed most of the CD45+ immune cells and those that remained were removed in the analyses. This is stated in the legend of **Fig. 3A** and also clarified this in text.

Question 4: Related to Figure 4

a. In lines 251-252, it states that “no significant changes in p16+/ppary- cells across the groups”. There are no images in Fig. 4 that support this statement.

b. In line 271, it states that “there were no changes in Ebf3-SPiDER+ cells in Veh and Doxo groups”. What about Ebf3-;tdTom+ cells?

c. Fig. 4M needs a clear label. Based on the results description, these μ CT images were taken from DTRADQ mouse treated with PBS or DT. PBS should be labeled above the first paired μ CT images, and DT instead of iDTRADQ should be labeled above the second paired μ CT images.

Response: In response to **question a**: we apologize for not initially referencing the figure number; the relevant data are now clearly indicated in **Fig. S6E**.

Regarding **question b**, we have quantified the **tdTom⁺ Ebf3⁻** cell population. Our analysis reveals an increased number of tdTom⁺ Ebf3⁻ cells in the Doxo group compared to the Veh group, suggesting that mature adipocytes also express tdTom. The quantification of **Ebf3⁻; tdTom⁺** cells have now been included in the revised figure **Fig. S6N**.

In response to **question c**, we apologize for the lack of proper labeling. We would like to clarify that both **DTR^{fff}** and **ADQ-Cre+/DTR^{fff}** mice were used in the study, and all mice received diphtheria toxin (DT) to assess potential off-target effects. We have now clearly indicated this in the images, main text, and figure legends.

Question 5: Related to Figure 5

In line 339 and Fig. 5C-F, please clarify which bone samples were used for RT-qPCR, bone marrow, crushed bone (depleted bone marrow), or whole bone.

Response: **Fig. 5C (now 6C)** corresponds to an IHC image, not to qPCR. **Fig. 5D-5F (now 6D-6F)** presents qPCR results obtained from bone where marrow was spun out (**in the revised manuscript, this sample is now referred to as the bone-resident fraction**) and the bones were then crushed and RNA was isolated. This approach is standard in the bone field as it allows for the elimination of the majority of CD45+ immune cells. While we acknowledge that some CAR cells/BMAd are removed by this approach, some remain as evidenced by the detection of CXCL12, FABP4, and PPARG mRNA (**Fig. 1A, shown below**). We would like to point out that these experiments were performed before we had discovered that CAR cells/BMAd senesced. For that reason, in **Fig. 6J**, we returned to our enrichment protocol where we digested whole bone, MACS depleted CD45+ cells and then used FACS sorting to isolate Ter119-, CD71-, CD31-, Sca1- and PDGFR β + cells, which significantly enriches CAR cells. Finally, **Fig. 6G-6I** is from BMAd that “floated” to the top of our digested bones, before we subjected the samples to MACs enrichment. Because mature BMAd are destroyed upon sorting, this was the most effective way to obtain mRNA from them.

Question 6: Related to Figure 6

a. Fig. 6M, IF staining of RANKL. Can the author include a description of the location of the stained RANKL in text? Is it primarily present in the bone marrow space or on the bone?

b. Line 412-413 states that the expression of other SNAP factors decreased after p38i/MK2i in crushed bone depleted bone marrow. Since senCAR/BMAds reside in bone marrow, expression of these SNAPs factors needs to be detected in bone marrow to further confirm the inhibitory effect of p38i/MK2i on SNAPs.

Response: In response to **question a**, we have now included a description in the text specifying the location of the stained RankL.

In response to **question b**, analysis of SASP expression was performed on samples where marrow was spun out to reduce the number of CD45+ cells and the residual marrow was crushed (**now referred to as the bone-resident fraction**) and used to isolate RNA. As mentioned above (response to point 5), a proportion of CAR cells/BMAds remain as evidence by the presence of CXCL12, FABP4 and PPARG mRNA (**Fig. 1A, above**).

Fig. 1A. RT-qPCR analysis of gene expression revealed enrichment of **CAR cells** and **BMAds** in the bone-resident fraction, as indicated by elevated Cxcl12, FABP4 and PPARG expression.

Question 7: Related to Figure 7

a. Representative images are needed for Fig. 7B

b. For Fig.7E-F. Bioluminescent images and intensity quantification are not sufficient to assess tumor burden. Please show histological images of tumor-bearing bones and quantify tumor burden by histomorphometry. Histomorphometric analysis is the gold standard for measuring tumor burden in bone.

c. Fig. 7G. How to distinguish bone loss caused by PyMT-BO-1 cells from paclitaxel (PTX)?

d. In Fig. 7J, tumors are not observed on the histological section, which is inconsistent with the bioluminescent images (showing tumor grew in bones). High-resolution images need to be provided in order to show tumor.

e. Given that (1) PTX and Doxo may have different anti-tumor effects on breast cancer, and (2) senolytic drugs (D+Q) may interact differently with chemotherapies, please provide justification for why PTX, not Doxo, was chosen to use in the breast cancer bone metastatic model.

The rationale for using the bone metastatic model to support the authors' findings is not clear. By introducing tumors in the model, it complicates the results and makes it hard to evaluate the changes in bone due to tumors or chemotherapy. The author may consider excluding the figure 7 from the manuscript since the model used here is not appropriate, and the results do not align with or support the main conclusion. Future work could investigate the author's findings by using adjuvant breast cancer models.

Response: In response to **question a**, we have added representative images for SA- β -gal staining, now included as **Fig. 8B**, and have clearly described these findings in the manuscript text.

For **question b**, we have added H & E stains of tumors to complement the BLI analyses BLI (**S15A**).

The reason we choose to use a metastatic model (**question c**) is that published data (Pin et. al., Cancer Lett 2022) showed that primary tumors drive bone loss by disrupting osteocytes. We piloted a primary tumor study by allowing the tumors to become palpable before the start of chemotherapy to more closely mimic the clinical scenario and allow us to measure tumor growth. However, according to this paper, by the time tumors are palpable, the mice would have already experienced bone loss. Using this model, we found that senolytics did not impact the anti-tumor efficacy of paclitaxel as expected. Unfortunately, but not surprising given this paper, we found only a trend that bone loss was rescued and we assume that is because tumor-induced bone loss is not senescence dependent. Given these findings we choose to test the metastatic setting and given we rescue the bone loss observed upon chemotherapy (**Fig. 8I**), we can separate the tumor versus chemotherapy-induced bone loss. That being said, we have added a sentence to point this caveat out in the discussion section when we discuss the implications of the data.

In response to **question d**, we have replaced the previous images with new ones that more clearly depict tumor-bearing bones (tumor areas are outlined with black circles, **Fig. 8J**).

Finally, for **question e**, the rationale for using paclitaxel (PTX) is that PTX is given as part of the neoadjuvant (or adjuvant) chemotherapy regimen for all breast cancer subtypes when chemotherapy is indicated. The use of Adriamycin (i.e., doxorubicin) is reserved for TNBC, or high-risk ER positive breast cancer and it is rarely used in HER2 positive disease. Because our tumor cell lines are luminal B, PTX would be the clinically relevant chemotherapy of choice. We have added a statement about this in the text to justify the use of PTX for these studies.

Question 8. Others

a. Detailed methods are needed on how the author defines and measures the OCN+ area on bone sections. **b.** In line 271, typo. Fig. 4SL should be Fig. S4L.

c. The phrase “bone resident cells” is used multiple times throughout the manuscript. The authors seem to use it to refer both to bone marrow cells and cells in/on bone, which causes confusion. Please clarify it in text or figure legend, as appropriate.

Response: In response to **question a**, we have added detailed information about the analysis of OCN⁺ cells at the end of the immunofluorescence staining methods section.

For **question b**, the typographical error has been corrected.

In response to **question c**, we again apologize for the confusion. We have gone through the manuscript and now refer to different fractions as follows:

1. **Bone resident enriched fraction** contains cells in which MACs kits were used to deplete CD45⁻; Ter119⁻; CD71⁻ cells.
2. **CAR cell enriched** refers to populations in which we have sorted for CD45⁻, Ter119⁻, CD31⁻, Sca1⁻ PDGFR β ⁺ cells.

3. **Bone-resident fraction** refers to crushed bone where marrow was spun out and bone resident and residual CD45+ cells remain. The presence of CAR cells/BMAds is ensured by the presence of CXCL12, FABP4 and PPARG mRNAs.

Reviewer #2 acknowledged that we used an extensive number of mouse models, genetic and pharmacological approaches to show that senescent bone resident CAR cells/BMAds contribute to bone loss through RANK ligand-driven osteoclast activation. In addition, the Reviewer stated that our study “is a well-presented and voluminous article in a hot topic of potential clinical relevance. The experiments are solid and elegantly designed, and the results align with the interest for a broad audience.” To strengthen the manuscript the reviewer made several suggestions that we address below.

Question 1: The authors must address the conceptual advance of this work for a journal like Nat Comm.

Response: This was addressed above.

Question 2: It is well known that osteocyte RANKL is required for cortical bone loss with age and that it can be induced by senescence (Kim et al. JCI insight 2020, PMID: 32870816).

Response: While we are aware of the paper, it should be noted that we do not see cortical bone loss in our system nor do we see senescent osteocytes. In addition, in the manuscript from Khosla and colleagues (PMID: 27341653), they found that in aging, osteocytes senesced and drove bone loss (there was no chemotherapy in this study). We would argue this supports our argument that identifying the senescent cell type, the stress that induced senescence, and the mechanism(s) by which it impacts biological processes is critical as each can have distinct affects. This is one reason our work is novel and provides important conceptual advancement.

Question 3. It seems concerning a recent publication from another laboratory (Ali et al. Scientific Reports 2025, PMID: 40389522) reporting that the adverse effects of chemotherapy on bone mass are not prevented by senolytics. This point needs to be convincingly addressed. How do the authors explain this apparent discrepancy?

Response: We appreciate the reviewer bringing this publication to our attention. The paper was published after we submitted our manuscript so at the time of submission, we were unaware of the work. The manuscript reports varied chemotherapy treatments (that differ from our dosing) that were delivered over several weeks to adult C57Bl/6 mice, which were of similar age to what we utilized in our studies. Interestingly, as we reported in our 2020 Cancer Research paper, after 4 weeks of treatment, we also failed to find senescent markers despite the fact that we could prevent bone loss using the INKATTAC mouse. Because we did not find senescence markers in that first publication, for this study we carried out a time course and dose course to find the window when senescent cells and the SASP were apparent, which was between 7 and 14 days. It is for that reason we choose the 9-day timeline in our current study. Furthermore, we find that our genetic and senolytic approaches prevent bone loss but fail to rescue bone that was already lost, which is what was reported in the Ali et. al., paper. We did not include this data in our manuscript because we cannot explain why we cannot rescue bone loss. Thus, we do not think our data is in conflict with the published paper, but instead distinct in that we are examining the effect of

senescence on initial bone loss. We have added this to our discussion and referenced Ali et. al.

Question 4.

Following with the previous point, a possibility raised by Ali and collaborators is that the decrease in osteoblast numbers is due to chemotherapy DDR-induced osteoblast apoptosis. Therefore, mechanistic analyses in this manuscript must include a comprehensive characterization of apoptosis (e.g. TUNEL or activated-caspase 3) damage makers (e.g. gH2AX) and how they relate to the crucial cellular types that maintain the intricate balance of bone damage/loss and formation/resorption.

Response: Per the reviewer's

suggestion, we performed immunofluorescence staining using osteocalcin (OCN) and TUNEL. Because DNA damage and subsequent repair and apoptosis can occur over time, we performed TUNEL staining at 1 hour, 2 hours, 6 hours, 24 hours, 72 hours and 9 days post-Doxo treatment. Our analysis revealed no significant difference in OCN⁺TUNEL⁺ signals between Veh- and Doxo-treated groups in any of the timepoints examined, indicating that apoptosis is unlikely to be the primary mechanism underlying the reduction in osteoblasts following chemotherapy (**Fig. 2A and 2B, shown above**).

Question 5. The authors use the well-established INK-ATTAC mouse model to provide evidence that chemotherapy-induced senescence associates with disrupted bone homeostasis, and also p16CreERT2-tdTom mice to identify CARs and BMADs with p16 "senescent" features. To ensure specificity of the p16 antibody the authors tested bones from p16KO treated with Doxorubicin. If the authors are correct in their hypothesis, I find crucial to validate the main conclusions of this manuscript in chemotherapy-treated p16KO mice. What would be the phenotype and mechanism of this model in bone loss and senescence induction?

Response: We have extensively used p16 to identify senescent cells and the p16 minimal promoter to drive our suicide gene (i.e. INK and now INK-QR mouse). We

Fig. 2. A. Immunofluorescence staining for **OCN** and **TUNEL** demonstrates that osteoblasts do not undergo apoptosis. **B.** Quantification of the **OCN⁺TUNEL⁺** area. Scale bar: 50 µm.

have also used senolytics to complement our studies and kill senescent cells regardless of p16 expression. We have no evidence that p16 expression itself is important for the phenotype we are studying. Furthermore, the relationship between p16 and SASP is complex. For example, Campisi and colleagues showed that overexpression of p16 itself induces a cell cycle arrest reminiscent of senescence but does **not** induce the SASP (PMID: 21880712). Further, shRNA directed reductions in p16 can reduce some SASP components but does not reduce expression to baseline (PMID: 33550279). Thus, p16's role in SASP is not straight forward and it is not clear what result we would obtain from the p16-KO mouse and how we would interpret it. Given the orthogonal mouse models we have already used including the newly added INK-QR mouse, our data indicate that bone resident CAR/BMAd drive bone loss.

Question 6. Also, the authors do not validate the potential role of p21 as senescence biomarker over the manuscript. In fact, the p53/p21 axis is an essential component of the chemotherapy-induced DDR response. Are senCARs and senBMAds positive for both p16 and p21? Or there are different senescent ecosystems? In this regard, district secretomes in p16 and p21 positive senescent cells have been proposed across tissues, including the bone (Saul et al. bioRxiv, <https://www.biorxiv.org/content/10.1101/2023.12.05.569858v1.full.pdf>).

Response: We appreciate the reviewer's comment and have provided evidence showing that p21 is elevated in bone marrow adipocytes (BMAds), as demonstrated in the original **Fig. 4E**. Additionally, we examined p21 (Cdkn1a) expression in our scRNA-seq data, specifically within the CAR cell population (**Fig. 3A, below**) and CAR cell-enriched sample (**Fig. 3B, below**) and observed a clear upregulation of p21 in response to Doxorubicin (Doxo). While we recognize that p21 plays an important role in senescence within the bone during aging, the key finding from our work remains that elimination of p16⁺ senescent CAR/BMAds is sufficient to prevent all of the bone loss, as shown in **Figures 1I and 2C**. Therefore, while p21 may have unique biology, it is unclear what additional insight would be gained by further validating p21 in our studies.

Question 7: Any validation with human datasets / samples post-chemotherapy regarding senCARs / senBMAds would reinforce the conclusions and translatability of this study, although I'm aware of the difficulties in getting such samples.

Response: As the reviewer points out, it would be difficult if not impossible to obtain these samples from solid tumor patients. Indeed, we have spoken with our clinical colleagues and they feel strongly that the IRB would have ethical concerns and therefore be unlikely to approve an invasive procedure (which this would be) that has no direct benefit to the patient. That being said, we were able to identify a dataset from bone marrow obtained from leukemia patients fifteen days after chemotherapy treatment and we find that p16 is expressed in CAR cells that lack MKi67 and GSEA analyses reveals a senescence signature. This has been added to **Fig. 3I and 3J** and strengthens our clinical relevance.

Fig. 3. A. scRNA-Seq dot plot shows elevated p21 expression in Doxo-treated CAR cells compared to Veh-treated CAR cells. **B.** RT-qPCR analysis confirms increased p21 expression in the Doxo-treated group relative to Veh in CAR cell-enriched samples.

Other points-The Discussion is currently underdeveloped and should address all concerns regarding novelty, contradictory data with other groups, and the limitations of this study.

- RNAseq analyses rely on SenMayo datasets, but it would be important to correlate the data with other datasets and senescent signatures.

Response: As stated above, at the time of submission we were unaware of the study by Ali et. al., and have now added it to our discussion as well as highlighting the importance of identifying the cell type senescing for the field.

In regard to the RNA-Seq analyses, it is increasingly recognized that the SASP and senescent transcriptomic signatures vary widely, depending on both the senescing cell type and the senescence-inducing stimulus (PMID: **28844647**). Many available signatures were generated by pooling data from diverse cell types or contexts, which may not capture the unique biology of bone-resident CAR/BMADs. Therefore, it is impossible to predict which if any signatures would overlap with ours. Thus, we argue that finding a senescence signature is sufficient and in line with other studies as a hypothesis driving exercise, which we confirmed experimentally, the ultimate support for our conclusions that adiponectin positive cells senesce in the bone and drive bone loss following chemotherapy.

Reviewer #3

Reviewer 3 felt that our manuscript described “a compendium of in vivo experiments using several genetic mouse lines, that chemotherapy-induced bone loss is linked to the induction of senescence specifically in adipo-lineage cells of the bone of

chemotherapy induced bone loss". However, the reviewer also raised some concerns that are addressed below.

Question 1: While depletion of adiponectin expressing cells or TNFSF11(RANKL) ablation specific to adiponectin expressing cells convincingly shows to prevent chemotherapy induced bone loss, the evidence of the senescence signature being restricted to CAR cells and adipocytes is less supported. The latter is mainly due to the usage of a variety of genetic mouse models or staining procedures which are not directly comparable (often using several methods to stratify the adipo-lineage but using a single approach to deny the role of other bone resident cells). This is surprising as the authors performed experiments, such as the adipocyte fractionation and scRNA-sequencing in which different compartments could have been compared using the same method to underline the susceptibility of adipocyte-lineage cells to senescence.

Response: We apologize as we think that our nomenclature for the cells used for different RNA analyses likely caused confusion. As we stated above, we have simplified the nomenclature throughout the manuscript. That being said, we would like to emphasize that our data demonstrate that the increase in p16+ expression and SA-B-gal is largely restricted to Ebf3+ cells (i.e., CAR cells), which are lost in INK mice treated with AP (**Fig. 4K and 4L**). However, to conclusively demonstrate that p16+ adiponectin positive cells are directly responsible for chemotherapy-induced bone loss and no other cells that we failed to capture with our orthogonal approaches, we created the ADQ-Cre/INK-QR mice that allow to activate the senescent suicide gene in a cell specific manner (**Fig. S8A, schematic**). Using these mice, we see loss of Ebf3+p16+ cells (**Fig. S5B and S8C**) and restoration of bone upon AP treatment (**Fig. 5C and 5D**).

Question 2: Trabecular bone loss has been initially characterized by loss of trabecular numbers; however, this parameter is not changed throughout the intervention studies in the INK-ATTAC mice, adipocyte depleted mice, mice lacking RANKL in adipocytes, or mice treated with senolytics or senomorphics. In addition, the effects of trabecular thickness are limited and therefore are unlikely to explain the observed bone loss. Did the authors check mineral content and density to explain the loss of trabecular bone without effects on trabecular numbers and thickness.

Response: We apologize for not supplying all of these parameters throughout. They were omitted in an effort to streamline the overwhelming amount of data. The BMD results are now presented here for your review (**Fig. 4A-4F, below**) and also added in the revised manuscript.

Question 3: The authors use a lot of space for the clustering the cell-type identification in the scRNA-seq data, but do not follow up on this throughout the manuscript. Therefore, clustering could be part of the supplement. In contrast, the gene set enrichment analysis should be performed by comparing Doxo condition to vehicle for each individual cluster to highlight that CAR cells are more susceptible to the induction of senescence than the other cells. For now, the authors compare CAR cells with the remaining cells, but not the induction levels in each cluster individually. A heatmap of the Sen-Mayo gene signatures could be additionally used to quantify changes in senescence associated gene transcription.

Response: We have moved the cell cluster markers plot to the Supplementary section (now presented as **Fig. S5G**). As suggested, we added a dot plot (**Fig. 3F**) that highlights senescence associated gene signatures that are upregulated in Doxo-treated cell clusters including CAR cells when compared to Veh-treated cells but not in other cell types. We believe this further emphasizes the impact of Doxo on CAR cells.

Question 4: Along the line, the fractionation of mature adipocytes for gene expression (Fig. 4D) should be complemented by analysis of the SVF fraction, i.e., the cells in the pellet, to compare induction levels of SASP factors across fractions.

Response: As the reviewer suggested, we obtained mRNA from the SVF and performed RT-qPCR and observed no change in **p16** expression in the Doxo group, whereas **p21** levels were notably elevated (**Fig. 5A and 5B, below**). The lack of p16 upregulation in the SVF is likely due to the fact that CAR cells are a minor population and was the reason after seeing our scRNA-Seq data, we moved back to an enrichment approach. Specifically, we performed qPCR on mRNA from the **CAR cell-enriched population** (CD45- Ter119- CD31- Sca1- PDGFR β +). In this refined population, we detected a marked increase in **p16** expression in the Doxo-treated group compared to Veh (**Fig. 4C**).

Question 5: While the authors emphasize to outline adipo-lineage cells to be highly susceptible to chemotherapy induced senescence and causally involved in bone loss, they do not show how adipocyte ablation and TNFSF11 conditional knockout are related to the senescence burden within the bone. Thus, estimation of senescent cells through P16 or SPIDER staining should be performed in the mice iDTR-ADQ mice to prove the loss of senescent cells as well as in TNFSF11-AdipoCRE mice to prove the persistence of senescent adipo-lineage cells.

Response: As per the reviewer's suggestion, we have performed immunofluorescence assays to identify senescent cells in bone sections of ADQ-Cre/iDTR, ADQ-Cre/Rankl, ADQ-Cre/INK-QR, and ADQ-Cre/MK2^{ff} mice. The results are now included in the revised manuscript and shown in **Fig. S7F and S7G** for the ADQ-Cre/iDTR model and in **Fig. S10M and S10N** for the ADQ-Cre/Rankl model. For ADQ-Cre/INK-QR, and ADQ-Cre/MK2^{ff} mice it is shown in (**Fig. S8B and S8C**) and (**Fig. S14B and S14C**) respectively. As shown in these figures, p16+ cells are lost in the ADQ-Cre/iDTR and ADQ-Cre/INK-QR mice while they are retained in the ADQ-Cre/Rankl mice and ADQ-Cre/MK2^{ff} mice, as expected.

Question 6: Data availability statement is lacking describing the deposition of raw as well as processed data. Making their data set accessible, especially a well-annotated Seurat object would have a big impact on the implementation of the data in future studies.

Response: The single cell data was deposited into GEO (GSE289491) at the time of submission and reviewers were granted access to the data. The data will be released upon publication to the public. In addition, we will upload our Seurat Objects to GEO so that they are also freely available. We apologize for failing to include the statement in the original submission.

Question 7: The panels characterizing the histological data, fluorescence as well as H&E or TRAP staining are often way too small.

Response: We can increase the size of the images but will need to increase the figure numbers or move additional data to the supplemental section. We ask for guidance on which approach is preferred.

Question 8: Quantification of osteoblasts (1M,S5D, 6I) should be done through measuring osteocalcin positive cells on the bone surface and not the total area of staining throughout the marrow space.

Response: We agree with the reviewer and would like to emphasize that for this quantification, OCN⁺ signals located within the marrow cavity were deliberately excluded from the analysis. Instead for quantification, OCN⁺ signals proximal to the bone surface were included. Importantly, the same analysis criteria and parameters were consistently applied across all treatment and control groups to ensure accuracy and comparability.

Question 9: The in vitro differentiation experiments are misleading, as the isolated cells of Doxo treated mice do not survive or do not proliferate to the extent of cells from vehicle treated mice. These data should rather be replaced with proliferation curves and CFU-assays but also be part of the supplementary figures.

Response: We thank the reviewer for their comment. The point of the in vitro assays was to show there were fewer cells capable of differentiation given CAR cells are

precursors for both osteoblasts and adipocytes. Because we plate the **same** number of living bone marrow cells per well (1×10^4), we believe these data support the conclusion that the precursors do not divide. As suggested, these data were moved to the supplementary figures.

Question 10: Is it correctly described that the gene expression analysis in Supplementary Figure 7J to L has been performed on crushed bones depleted for bone marrow? Wouldn't this imply that the material is devoid of both CAR cells and adipocytes, but still the authors find induction of SASP members? How does this relate to the fact that senescence is restricted to adipo-lineage cells?

Response: As we responded to reviewer 1, we apologize for the confusion created by our nomenclature. For these experiments, marrow was spun out but some CAR/BMAd remains as evidenced by detection of CXCL12, FABP4 and PPARG mRNA (**Fig. 1A**).

Question 11: The authors should implement a time line for the diphtheria toxin and tamoxifen treatments, i.e., how many days after adipocyte or RANKL ablation die, they initiate Doxo or PTX treatment. This would help to understand the timing to establish the basal differences in the iDTR-ADQ and TNFSF11-AdipoCRE mice to their respective controls.

Response: As per reviewer suggestion, we have added timelines for ADQ-Cre/DTR and ADQ-Cre/Rankl in **Fig. S7A** and **Fig. S10H**, respectively. Timelines are usually located in the supplementary figure due to space but also described in the method section.

Question 12: Color scale in Fig3G is not optimal, since the CAR cells 3 could have a huge white circle for Cdkn2a which is not visible. Since the color bar indicates increasing levels, making the color matrix going through white is not optimal.

Response: Based on the suggestion, we have changed the color bar to improve visibility. The new scale goes from red to yellow to blue, without any white, so that even low expression levels would be visible. This dot blot shows that Cdkn2a in CAR cells 3 is not expressed (see small, blue dot). To emphasize this point, we also increased the dot size scale (from 30 to 60), which made the dot slightly more visible. Despite these changes, the figure shows that Cdkn2a is not expressed in this population. As requested, the image has been added to the revised manuscript (**Fig. 3H**).

Question 13: The authors should state the house keeping gene in the figure panels as the method section indicates that different ones have been used.

Response: This information has been added to the figure legend as per the reviewer's comments.

Question 14: The TNFSF11 induction in the scRNA-seq data should be rephrased, as the majority of CAR cells show down-regulation of TNFSF11 while a new population of cells with very high expression levels appear. In addition, a summed score of the Sen-Mayo gene signature could be used to correlate TNFSF11 expression levels. The latter can demonstrate that CAR cells with this very high TNFSF11 expression levels are also high in senescence-related gene expression.

Response: We have visualized Tnfsf11 expression in CAR cells 1 using a dot plot, which clearly illustrates the emergence of a subpopulation with markedly high expression following Doxo treatment (**Fig. S10G**). Furthermore, we found that

RANKL⁺ senescent CAR cells exhibit a pronounced senescence-associated transcriptional profile, as demonstrated by the expression of p21 (Cdkn1a) in the dot plot and additional senescence-related genes (**S10G**). These results indicate that the highly RANKL-expressing CAR subpopulation overlaps with cells displaying elevated senescence signatures.

Question 15: The authors could add a statement on the limitation of the study, i.e., not being able to de-tangle the effect of CAR cells and mature adipocytes. In addition, the study aims to prevent bone loss but did not test whether senolytics or interference with adipocyte derived RANKL expression after established bone loss can restore bone over time.

Response: We have added a sentence to the discussion that our data does not establish whether it is CAR cells or adipocytes that are dominant (or equivalent) in bone loss. Further, as indicated in response to Reviewer 1, we have data that suggests once lost, bone cannot be rescued by eliminating senescent cells.

Question 16: The observation that TNFSF11-AdipoCRE mice have higher bone mass and loss of osteoclast on trabecular bone surfaces raises the question whether adipo-lineage derived RANKL is important for normal bone homeostasis and not only under pathological conditions, as the authors refer to (line 342). How is site-specificity obtained in a model in which cells distant to the bone surface control osteoclast maturation?

Response: We thank the reviewer for the comment. It is true the baseline bone mass is higher in the TNFSF11-adipoq-Cre model as was reported by others (PMID: 34121311), so RANKL is important in normal bone homeostasis as well as pathological conditions. We have added a statement to the discussion about this as requested. That being said, despite the increased bone mass, loss of RankL in adiponectin positive cells make the mice resistant to chemotherapy-induced bone loss. To bolster our conclusions, as we stated above, we now have data in the ADQ-Cre/INK-QR mouse that does not show increased baseline bone mass. Using this model, we can also prevent chemotherapy-induced bone loss so we feel this addresses the concern. This is now presented as (**Fig. 5C and 5D**).

Question 11: If not addressing in the future, the authors should discuss whether baseline adiposity in the marrow cavity affects chemotherapy induced bone loss, and if site-specific bone loss corresponds to local adipo-lineage content.

Response: We agree this is an important question but feel it is outside the scope of this study, and an important subject for future investigation.

Question 17: Finally, there is a trend of inconsistency of how to convey the main message, exemplified by the senolytics and senomorphics treatment. In the first senescence is estimated through the SPIDER assay and co-staining for EBF3 to focus on CAR cells, while with senomorphics the authors show P16 staining. The first intervention is supported by osteoblast numbers while the latter is through bone formation assays. The effect on RANKL levels are only documented for the senomorphics. This style appeals as existing data have been compiled. For that I would suggest to focus on common measures in the main figures, but this is clearly a question of taste.

Response: We apologize the inconsistency in staining but would like to clarify the reason for the different approaches. When we are looking for loss of p16⁺ cells (i.e., the INK-ATTAC mice or ADQ-Cre/DTR or ADQ-Cre/INK-QR mice), we use an antibody

for p16, which is from the same species as the Ebf3 antibody. Thus, we cannot use the antibodies for p16 and Ebf3 together. When we wish to show senescent Ebf3 positive cells are lost, we use co-staining of SPiDER- β -Gal and Ebf3.

Regarding osteoblast numbers/activity. We first examined changes in osteoblasts by measuring OCN+ cells in the INK mice and following senolytics. For the senomorphics, which may also translate to the clinic, we wanted to ask how they also impacted osteoblast activity. Finally, we have added RANKL measurements for all experimental approaches where we isolated bone resident enriched cells.

Please note, major changes/additions to the text are highlighted in light grey.

Below we provide a point-by-point response to each reviewer comment.

REVIEWER#1

The authors have addressed most of my concerns thoroughly. The quality of results related to the bone metastatic model (Figure 8) has been improved. I thank the author for adding the histomorphology assessments to support their findings.

We thank the reviewer for this comment.

And, justify the reason for choosing the bone metastatic model over the primary tumor model. In fact, I think a non-metastatic breast cancer model is more suitable for testing their hypothesis than a bone metastatic breast cancer model, since it is tricky to distinguish bone destruction caused by chemotherapy or metastatic cancer. Although the author mentions a paper (Pin et. al., Cancer Lett 2021) states that primary tumors can also cause bone loss through systemic endocrine effects, the results from the paper were not obtained using breast cancer model.

We have added a discussion about why we used the metastatic model to test our therapeutic approach and have added the Pin reference to our manuscript. As we stated in our discussion, give primary tumor growth can cause bone loss we did not wish to confound our findings with that approach.

REVIEWER#2

The authors have now sufficiently addressed my concerns. I have no further comments and congratulate the authors for the work done..
We thank the reviewer for their helpful comments.

REVIEWER#3

Ganes Kumar Raut et al. did a great job in answering the technical as well as conceptual concerns raised during the revision of the manuscript. Importantly, in the revised manuscript the authors made use of their existing data to show that Car cells are especially susceptible to the induction of a senescence gene signature along with induction of Tnfsf11 expression. In addition, they also follow up on the presence of senescent cells when interfering with adipocyte numbers or Tnfsf11 gene expression.

We thank the reviewer for their helpful comments.

Major:

The main finding of the study only provides limited advantage to the field. As outlined in the rebuttal letter, the mechanistic insight is given by the identification of CAR and adipo-

lineage cells as main cell type being responsible for chemotherapy-induced bone loss through a senescence-driven induction of RANKL production. While the authors argue with an important statement of: “These results reveal that even within the same tissue (i.e., the bone), the nature of the insult determines the cell type that senesces and the downstream consequences, highlighting, not diminishing the broader biological and clinical significance of our findings. Together, our findings and Khosla’s work set a standard for the field by saying that simply showing that phenotypes get better upon the elimination of senescent cells without identifying the cell type or mechanism responsible is no longer sufficient.” One needs to be aware that the induction of senescence specifically in bone marrow adipocytes is linked to glucocorticoid-induced bone loss (<https://doi.org/10.1016/j.cmet.2023.03.005>). The paper by Liu et al. also showed that transfer and elimination of senescent adipocytes can induce and rescue bone loss, respectively, already demonstrating the compartmentalization of senescence within bone tissue.

We thank the reviewer for pointing out this paper and while it is clear that glucocorticoids can induce bone loss upon transfer, in our setting, we show that it is RANK ligand derived from endogenous senescent CAR cells that drives bone loss in response to chemotherapy.

Despite its potential clinical relevance, and the clear and thorough execution of experiments, I judge the conceptual novelty to be limited. As such, conceptual novelty would be the identification of molecular cues driving the elevated susceptibility of CAR cells and bone marrow adipocytes to chemotherapy induced senescence to understand compartmentalization of senescence within the bone tissue. Or how senescence itself triggers RANKL expression and whether this is distinct from Tnsf11 induction during normal bone remodelling or auto-immune induced bone loss.

We thank the reviewer for their opinion and show that it is the activation of the p38MAPK-MK2 pathway that leads to expression of Rank ligand in our setting. As to why CAR cells are specifically senescing, given the extensive work we have done identifying the senescent cell in the bone and how it drives bone loss, we argue that determining why CAR cells specifically senesce is beyond the scope of our current manuscript. Importantly, I recognize the strength of the data and rigor of the presented study, and I feel comfortable with the editor’s decision to consider this manuscript for publication in nature communications.

We thank the reviewer for their acknowledgement of our work.

Minor:

Conceptually, the in vitro differentiation data do neither support the main story line nor justify the conclusion taken in the manuscript of an impaired osteoblast differentiation capacity if cell numbers are the primary problem. As outlined in the revision, the authors claim that their data show impairment in proliferation, which is neither the conclusion in the main text nor supported by the assays as increased cell death could have a similar outcome. Those data might be removed from the manuscript.

We agree that we cannot conclusively conclude if we are seeing reduced proliferation and/or apoptosis so simply report the impaired differentiation. Per the editors request, we have left the data in the manuscript.

The figures have random styles of colours and shapes, which does not compromise the scientific value or message, but might lead to confusion if use of colours is intended. Not limited to, but for example Fig 2C versus Fig 2F/2G and S3A-E,M or O for the INK-ATTAC mice or Fig 5B versus S7M for the ADQ-CRE/DTR mice.

We have gone back and attempted to standardize the styles and colors.

Line 314, it should refer to Figure 5C and 5D instead of 4C and 4D. We have corrected this and thank the reviewer for catching it.

REVIEWER#4

We thank the reviewer for their helpful comments.

REVIEWER#5

We thank the reviewer for their helpful comments.